# Virion morphology and on-virus spike protein structures of diverse SARS-CoV-2 variants

Zunlong Ke [1,2,3], Thomas P Peacock [4,5], Jonathan C Brown [4], Carol M Sheppard [4], Tristan I Croll[6,7], Abhay Kotecha[8], Daniel H Goldhill [4,9], Wendy S Barclay[4] & John A G Briggs [1,2]✉

## Abstract

**The evolution of SARS-CoV-2 variants with increased fitness has been accompanied by structural changes in the spike (S) proteins, which are the major target for the adaptive immune response. Single-particle cryo-EM analysis of soluble S protein from SARS-CoV-2 variants has revealed this structural adaptation at high resolution. The analysis of S trimers in situ on intact virions has the potential to provide more functionally relevant insights into S structure and virion morphology. Here, we characterized B.1, Alpha, Beta, Gamma, Delta, Kappa, and Mu variants by cryo-electron microscopy and tomography, assessing S cleavage, virion morphology, S incorporation, "in-situ" high-resolution S structures, and the range of S conformational states. We found no evidence for adaptive changes in virion morphology, but describe multiple different positions in the S protein where amino acid changes alter local protein structure. Taken together, our data are consistent with a model where amino acid changes at multiple positions from the top to the base of the spike cause structural changes that can modulate the conformational dynamics of the S protein.**

**Keywords** Coronavirus; Cryo-electron Tomography; Membrane Fusion Protein; Virus Evolution; Virus Structure
**Subject Categories** Microbiology, Virology & Host Pathogen Interaction; Structural Biology

## Introduction

The accumulation of genetic variation in SARS-CoV-2 spike (S) protein has had striking effects on transmission, antibody resistance, pathogenesis, and diseases (Harvey et al, 2021). The start of the pandemic was driven by a lineage B virus hereafter refer to as the index strain (Zhou et al, 2020a). Early in the pandemic, the S protein obtained an aspartic acid (D) to glycine (G) mutation at position 614 which corresponded with an increase in

transmissibility (Volz et al, 2021). The most widespread lineage with D614G (lineage B.1) became the dominant genotype globally (Plante et al, 2021). Over the course of 2020, B.1 was replaced by genetically similar direct ancestors, before being overtaken by several lineages towards the end of 2020/start of 2021, referred to as variants of concern (VOCs) or variants of interest (VOIs). Different VOCs/VOIs arose and become dominant in different regions, with the Alpha (B.1.1.7) (Liu et al, 2022b) variant in Europe, the Beta variant (B.1.351) in Southern Africa, and the Gamma (P.1) and Mu (B.1.621) variants across South America. In early 2021 the Delta variant (B.1.617.2) arose in India (alongside the B.1.617.1 Kappa variant). The Delta variant then become the dominant global lineage outcompeting all previous VOCs/VOIs. Towards the end of 2021, the Omicron variant (BA.1) carrying over 30 mutations in S became the major variant circulating globally (Viana et al, 2022). As of December 2023, relatives of those lineages, still classified as Omicron lineages, continue to circulate as the predominant lineages worldwide with few exceptions (Roemer et al, 2023).

SARS-CoV-2 is an enveloped positive-sense RNA virus in the family of *Coronaviridae*. The 30 kb linear RNA genome is encapsidated by the ribonucleoprotein complex (RNP) forming a "beads-on-a-string" structure (Yao et al, 2020). The viral genome encodes 4 structural proteins: S, membrane (M), envelope (E), and nucleocapsid (N) proteins. S glycoprotein trimers protrude out from the viral membrane and are the major proteins on the viral surface, making it the main target for vaccine design. Early SARS-CoV-2 strain S harbored a suboptimal polybasic cleavage site at the S1/S2 junction ([681]PRRAR[685]), that allowed cleavage by cellular furin. This polybasic cleavage site appears absent in other closely related sarbecoviruses. The presence of this polybasic cleavage site results in cleavage of S during glycoprotein trafficking (Guo et al, 2022) and was shown to be a key determinant for virus replication, pathogenesis, and transmissibility (Hoffmann et al, 2020; Johnson et al, 2021; Peacock et al, 2021a). Sequential cleavage of S at the S1/S2 site followed by the S2' site within the S2 domain exposes the fusion peptide (FP), which mediates host-viral membrane fusion, allowing entry of the viral genome into the host cell. It has been reported that a highly flexible fusion-peptide proximal region (FPPR), downstream of the FP, contributes to regulation of the conformational states of S (Zhang et al, 2021a). Several VOCs/VOIs

[1]Department of Cell and Virus Structure, Max Planck Institute of Biochemistry, Martinsried, Germany. [2]Structural Studies Division, Medical Research Council Laboratory of Molecular Biology, Cambridge, UK. [3]Department of Molecular Biosciences, The University of Texas at Austin, Austin, TX, USA. [4]Department of Infectious Disease, Imperial College London, London, UK. [5]The Pirbright Institute, Woking, UK. [6]Cambridge Institute for Medical Research, University of Cambridge, Cambridge, UK. [7]Altos Labs, Cambridge, UK. [8]Materials and Structural Analysis, Thermo Fisher Scientific, Eindhoven, The Netherlands. [9]Department of Pathobiology and Population Sciences, Royal Veterinary College, London, UK. ✉E-mail: briggs@biochem.mpg.de

have convergently evolved substitutions proximal to the S1/S2 cleavage site at P681 that enhance furin cleavage, such as P681H (Alpha, Mu, and Omicron) and P681R (Delta, Kappa, and Omicron BA.2.86) (Liu et al, 2022a; Khan et al, 2023). Enhanced cleavage of the full-length S into S1 and S2 is thought to lead to higher efficiency in the viral entry process and virus transmissibility (Carabelli et al, 2023; Liu et al, 2022a).

In vitro structural characterization of recombinant, purified S by cryo-electron microscopy (cryo-EM) is a powerful tool to study this fusion machinery. It allows understanding of the molecular architecture of S (Wrapp et al, 2020; Walls et al, 2020; Xiong et al, 2020; Toelzer et al, 2020; Cai et al, 2020; Henderson et al, 2020), it offers possible structural explanations for the interaction between S and host receptors (Yan et al, 2020; Zhu et al, 2021; Zhou et al, 2020b), and it provides insights into viral immune evasion via interactions between S and antibodies (Barnes et al, 2020; McCallum et al, 2021c, 2021b; Cerutti et al, 2021). Taking advantage of this established method, cryo-EM has been used to determine structures of recombinant, purified S trimers from emerging variants including Alpha, Beta, and Gamma (Cai et al, 2021; Yang et al, 2021; Gobeil et al, 2021; Mannar et al, 2022a), as well as Delta, Kappa, and Epsilon (Zhang et al, 2021b; McCallum et al, 2021c; Saville et al, 2022; McCallum et al, 2021a) and, more recently, Omicron (Mannar et al, 2022b; McCallum et al, 2022; Yin et al, 2022; Hong et al, 2022). These studies offer possible structural explanations for how mutations in S could contribute to the observed transmission and immune evasion phenotypes (Jackson et al, 2022), often by modulating immunogenic sites. These studies have also described differences in the range of conformational states of the N-terminal domain (NTD) and receptor binding domain (RBD), which can move between open, closed, and locked states (Xiong et al, 2020; Qu et al, 2022; Toelzer et al, 2020; Walls et al, 2020; Henderson et al, 2020). A direct comparison between different studies can be difficult, due to differences in the techniques used to prepare the recombinant protein, including the introduction of stabilizing mutations, different producer cell lines, lipid environments, and purification methods.

In situ structural studies by cryo-electron tomography (cryo-ET) of SARS-CoV-2 assembly and isolated virions reveal the viral morphology, 3D organization of structural proteins, and the distribution and conformational states of S on virions (Klein et al, 2020; Ke et al, 2020; Turoňová et al, 2020; Yao et al, 2020; Liu et al, 2020; Mendonça et al, 2021). These earlier findings indicate that the virions are approximately spherical with a diameter of 80–90 nm and 25–48 S per virion on the B.1 strain (Ke et al, 2020; Klein et al, 2020; Turoňová et al, 2020; Yao et al, 2020). Here also, the techniques used to prepare the viral samples may influence the conformational state distribution of S, including the ratio of prefusion and postfusion forms. Preparation methods may also influence virion morphology, and recently, cryo-ET studies of emerging variants suggested some may have non-spherical viral morphology, such as cylinder-shaped virions from the index strain and three emerging variants (Calder et al, 2022) and membrane-invaginated virions from the Delta variant (Song et al, 2023).

Here, we describe virion morphology, and perform in situ structure determination of S from a panel of SARS-CoV-2 emerging variants including B.1, Alpha, Gamma, Delta, Kappa, and Mu. By imaging inactivated virions in their close-to-native state by cryo-EM and cryo-ET, we have explored how emerging variants differ in virion morphology, S density, and the ratio of pre- and post-fusion conformation. Then, we resolve the high-resolution structures of S from the native virions, evaluate their structural differences, identify key structural changes and assess the range of observed conformations. Our study offers a systematic characterization of emerging variants as a basis for further understanding of the evolution of SARS-CoV-2.

# Results and discussion

## Cleavage efficiency differs between variants

The amino acid changes in the variants studied here, as well as an evolutionary tree of the variants, are summarized in Fig. EV1A–C. To assess the effect of these amino acid changes on the cleavage efficiency of the variants, virions were harvested from infected Vero cells 72 h post infection, and split into two pools which were either inactivated with 4% paraformaldehyde and concentrated by ultracentrifugation for subsequent electron microscopy, or were concentrated using protein concentration columns, inactivated by heating in Laemmli buffer and subjected to western blot using anti-S2 and anti-nucleoprotein antibodies (Peacock et al, 2021a) (Fig. 1A). We quantified the amount of cleaved S2 and uncleaved S (S0) via densitometry analysis and calculated the cleavage efficiency as the ratio $S2/(S0 + S2)$ (Fig. 1B). This analysis suggests large variations in furin cleavage efficiency among the variants.

Mutations directly adjacent to the S1/S2 furin-mediated cleavage site have been shown previously to affect cleavage efficiency (Liu et al, 2022a; Brown et al, 2021). Gamma (P681), retaining the proline at position 681, shows cleavage at levels similar to B.1. Alpha and Mu have a P681H mutation ([681]PRRAR[685] to [681]HRRAR[685]) at the furin cleavage site, while Delta has a P681R mutation, making the furin cleavage site [681]RRRAR[685]. The efficiency of furin-mediated processing can be ordered: P681R (Delta) > P681H (Alpha and Mu) > P681 (B.1 and Gamma), in broad agreement with other studies using authentic live virus particles (Saito et al, 2022; Peacock et al, 2021a) and viruses generated using reverse genetics (Liu et al, 2022a). The fact that the furin cleavage efficiency correlates well with the amino acid at position 681 implies that other amino acid changes elsewhere in S have only marginal effect on the furin-mediated processing efficiency (Fig. 1). Data using variant-matched pseudoviruses showed a strong correlation with the live virus data with pseudoviruses containing P681R and P681H showing a higher amount of cleavage than those without any changes in the S1/S2 site (Fig. 1C,D). Although total magnitudes of cleavage differed between pseudovirus and virus-expressed S, this is consistent with previous observations, and could be due to differences in how S is expressed and the cell types it was expressed in.

## Insights into the mechanism of enhanced cleavage by SARS-CoV-2 variants

Several different hypotheses have been proposed to explain why mutations at position 681 influence spike cleavage. These include (i) direct changes in the S1/S2 loop charge (through the addition of positively charged amino acids arginine or histidine), (ii) removal of the inflexible proline 681 residue allowing greater flexibility of

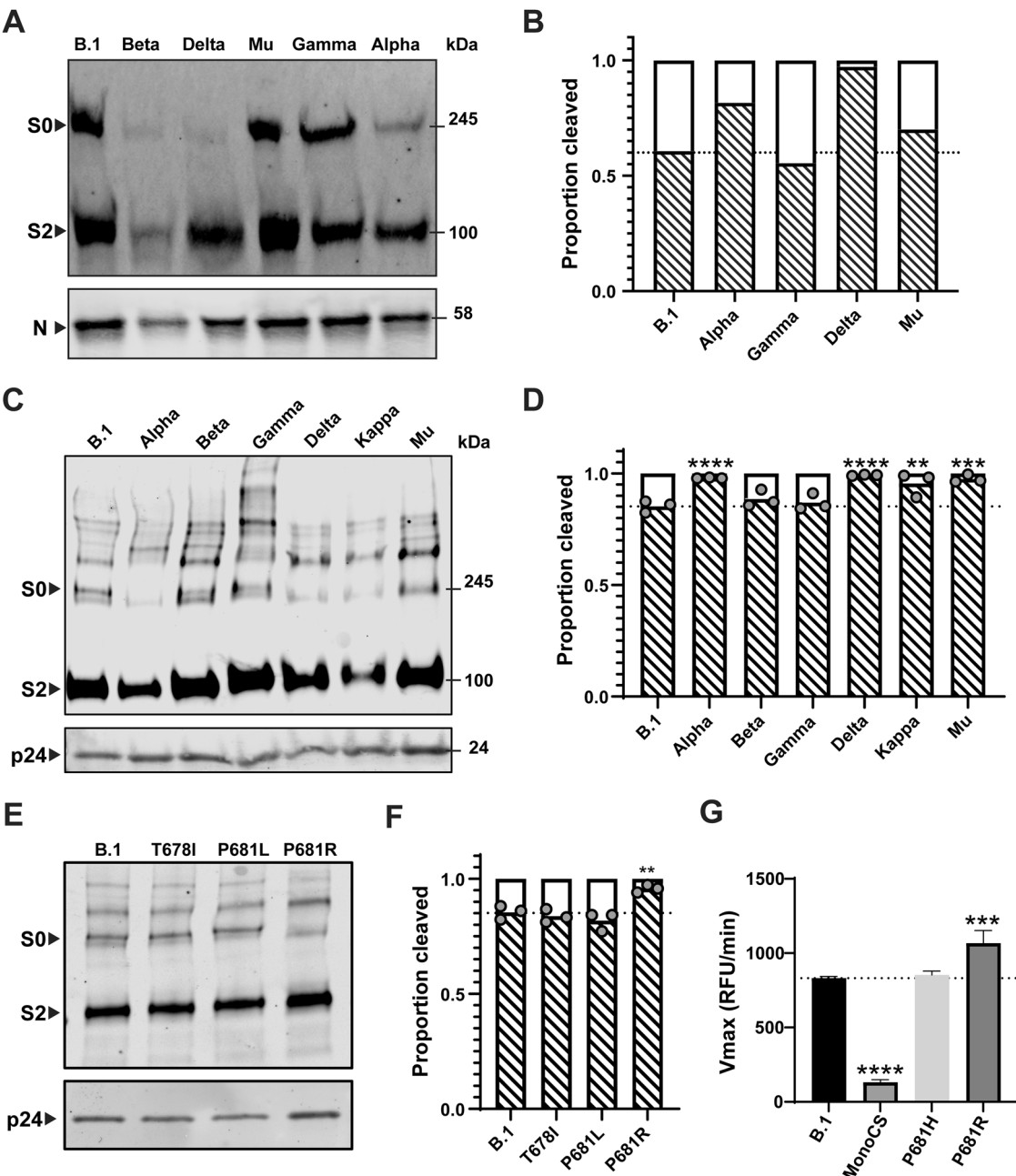

**Figure 1. SARS-CoV-2 S characterization among variants.**

(A, B) Virus cleavage patterns of the matched viruses used for cryo-EM analysis. (A) Western blot analysis indicates spike and nucleoprotein (N). S0 indicates uncleaved S and S2 is a cleaved product of S. (B) Densitometry quantification of S2/(S0 + S2) ratios. Beta variant was not quantified due to insufficient virus in the preparation. (C–F) Pseudovirus cleavage patterns of the matched pseudoviruses to the variants used in this study with (C, E) western blot analysis (representative blots shown) and (D, F) quantification using densitometry of three individual repeats ($N = 3$). B.1 vs Alpha, $P < 0.0001$; B.1 vs Beta, $P = 0.601$; B.1 vs Gamma, $P = 0.976$; B.1 vs Delta, $P < 0.0001$; B.1 vs Kappa, $P = 0.0011$; B.1 vs Mu, $P = 0.0001$. B.1 vs T678I, $P = 0.824$; B.1 vs P681L, $P = 0.379$; B.1 vs P681R, $P = 0.008$. (C, D) show whole variant spikes while (E, F) show glycosylation mutants. HIV p24 used as a loading control throughout. (G) Cleavage of SARS-CoV-2 spike S1/S2 fluorogenic peptide mimetics by recombinant furin, plotted as maximum enzymatic activity (Vmax). All assays performed in technical triplicates ($N = 3$) with a representative repeat from three completely independent repeats ($N = 3$) shown. Graph plotted as mean + Standard deviation. WT vs MonoCS, $P < 0.0001$; WT vs P681H, $P = 0.886$; WT vs P681R, $P = 0.0006$. Statistics throughout this figure performed by one-way ANOVA with multiple comparisons against B.1 plotted on the graph. **$0.01 \geq P > 0.001$; ***$0.001 \geq P > 0.0001$; ****$P \leq 0.0001$. Source data are available online for this figure.

the S1/S2 loop and better cleavage by furin, or (iii) removal of proline 681 results in a lower proportion of O-linked glycosylation at the nearby position 678, resulting in greater accessibility of the S1/S2 loop to furin (Gonzalez-Rodriguez et al, 2023; Lubinski et al, 2022a; Zhang et al, 2021c).

To test these different hypotheses, we generated and tested cleavage of pseudovirus containing spike mutations T678I (which would ablate O-linked glycosylation) and P681L (which would remove the proline, induce greater flexibility and reduce O-linked glycosylation without changing the charge of the S1/S2 loop). Neither of these mutants resulted in enhanced S1/S2 cleavage in our system (Fig. 1E,F), suggesting that the enhanced spike cleavage seen in P681H/R variants is most consistent with increased positive charge in the S1/S2 loop.

Furthermore, we tested the ability of recombinant furin to cleave fluorogenic peptide analogs of the S1/S2 loop. These peptides contained either the wild-type S1/S2 loop (from lineage B.1), a monobasic mutant version, or mutations equivalent to P681H or P681R. As expected, the B.1 peptide was efficiently cleaved by recombinant furin while the monobasic mutant was not (Fig. 1G). Consistent with previous publications, P681R showed a significant enhancement of furin cleavage, while the P681H mutant did not (Lubinski et al, 2022a, 2022b). This is in contrast to our previous pseudovirus data which suggested P681H alone enhanced cleavage (Peacock et al, 2022). The reason for this discrepancy between the pseudovirus, live virus and peptide data remains unclear, but could be due to peptides not quite capturing the conformation of the S1/S2 loop, or a difference in cleavage conditions within the peptide cleavage buffer versus the ER-Golgi intermediate compartment of the cell.

## Variants all have spherical morphology but differ in S incorporation

Virus-containing supernatant from infected cells, inactivated with 4% PFA, was clarified to remove cellular debris, concentrated, plunge-frozen, and imaged by cryo-EM. The Beta variant generated a low concentration of virus, as assessed by western blot (Fig. 1A) and subsequent cryo-EM images, and was excluded from further analysis. Virion morphology was assessed with a combination of 2D projections (cryo-EM) and 3D reconstructions (cryo-ET) (Fig. 2A,B). For all variants, the majority of virions were approximately spherical. A small fraction of "dented" virions (with regions with negative membrane curvature) was observed: 38% for B.1 ($n = 72$) and less than 15% for other variants ($n > 69$). Reexamining a subset of our previous cryo-ET data of B.1 virions in cell supernatant prior to sucrose-cushion ultracentrifugation (Ke et al, 2020), we found all virions ($n = 57$) to be spherical without any membrane invagination or deformation. The observed spherical morphology is consistent with earlier studies of B.1 strains and the index strain from us and others (Ke et al, 2020; Turoňová et al, 2020; Yao et al, 2020). However, it contradicts recent studies suggesting that Delta virions contain membrane invaginations (Song et al, 2023) and that the index strain and three variants (Alpha, Beta, and Delta) have a more cylindrical morphology (Calder et al, 2022). We consider it likely that the appearance of unusual morphologies is due to sample preparation, and that SARS-CoV-2 variants are generally spherical.

We measured virion diameter from both 2D projections and 3D tomographic reconstructions (Figs. 2C,D and EV2A–C). Virions have diameters between approximately 60 nm and 120 nm, with a mean diameter of around 80 nm (Fig. EV2). Overall, the measured virion diameters agree with previous studies of virions from B.1 and index strains before and after concentration (Ke et al, 2020; Yao et al, 2020; Turoňová et al, 2020). The mean virion diameter is slightly larger for B.1 than for other variants, but the small difference may reflect inherent variability in sample preparation or cell health (Fig. EV2).

We observed abundant S trimers protrude out from the viral membrane, forming the characteristic coronavirus "crown" around the virion on both 2D projections and 3D tomographic reconstructions. Ribonucleoprotein (RNP) density was observed in all variants (Fig. 2B). From 3D tomographic volumes, we quantified the number of prefusion and postfusion S trimers for each variant ($n = 15$) by visual examination (Fig. 2B). Delta ($34 \pm 14$) and Mu ($35 \pm 16$) had more prefusion S trimers per virion than B.1 ($20 \pm 10$); Alpha ($23 \pm 12$) and Gamma ($27 \pm 13$) had similar numbers of prefusion S trimers as B.1 (Fig. 2C). The number of postfusion S trimers in each virion were similar with only Mu ($2.8 \pm 2.2$) showing more postfusion trimers than B.1 ($1.3 \pm 1.5$). Taking the virion diameter into account, the density of S trimers (per 1000 nm$^2$ surface area) in Gamma ($1.5 \pm 0.6$), Delta ($1.9 \pm 0.9$), and Mu ($1.9 \pm 0.7$) is higher than in B.1. S-trimer density on Alpha ($1.3 \pm 0.6$) is similar to B.1 ($1.0 \pm 0.5$) (Fig. 2D) contrasting with Alpha variant lentiviral pseudoviruses which have been reported to have much lower S incorporation than B.1 pseudoviruses (Niemeyer et al, 2022). The measured density of trimers in B.1 is very similar to that measured from the almost identical strain used in our previous study (Ke et al, 2020) (Fig. 2D). While it is tempting to speculate that more recent strains have evolved higher densities of S on the viral surface since each variant was only prepared and imaged once in this study, we cannot rule out that these differences reflect variation in incorporation independent of strain.

## High-resolution structures of S trimers in situ

To resolve high-resolution structures of S trimers in situ from virus particles of different SARS-CoV-2 variants, we collected cryo-EM datasets, automatically picked S trimers on the periphery of the virions, and determined structures using single-particle cryo-EM processing pipelines in RELION as previously described in Ke et al (Ke et al, 2020). After a series of image processing steps, we obtained high-resolution density maps of S trimer from B.1 (2.7 Å), Alpha (3.2 Å), Gamma (3.2 Å), Delta (3.3 Å), and Mu (2.8 Å) (Appendix Fig. S1; Appendix Table S1). Overall, the S-trimer structures are similar to those previously determined for B.1 in situ, and are similar across all variants. The membrane-proximal region is not well-resolved in all S structures; this is due to its inherent flexibility as revealed by earlier studies (Ke et al, 2020; Turoňová et al, 2020; Yao et al, 2020).

Based on the high-resolution maps, atomic models were built for each variant. Mapping the amino acid mutations onto the S structure highlights that the residue changes are predominantly towards the periphery of S1, particularly in the NTD and RBD domains (Fig. 3A). In order to detect structural changes at the residue level, we aligned the S structure from each variant to B.1, and calculated the root-mean-squared deviation (RMSD) at the position of the alpha carbon (Cα) for each amino acid (Fig. 3B). Cα RMSD could not be calculated for unresolved, flexible regions in the NTD and RBD domains. High Cα RMSD values mark local structural changes between the variants and B.1. Between Alpha and B.1, the 570 hairpin (residues 565–575), 630 loop (residues

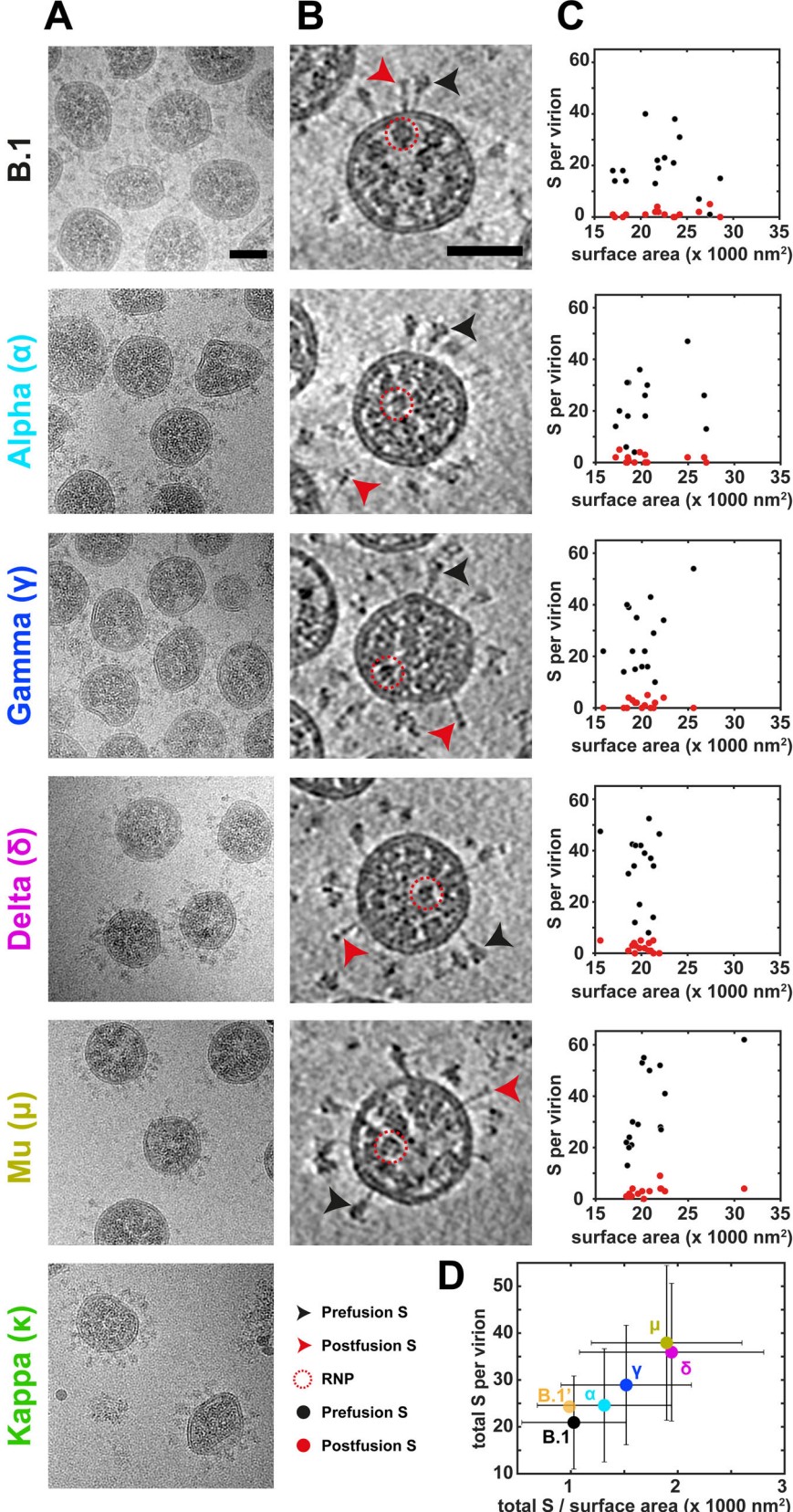

**Figure 2. Virion morphology characterization and S incorporation among variants.**

(A, B) Virion morphology as observed in 2D projections obtained by cryo-EM (A) and in slices through 3D reconstructions obtained by cryo-ET (B). (B) Representative prefusion S (black arrowhead), postfusion S (red arrowhead), and ribonucleoprotein (RNP) (dotted circle) are indicated, see key. The scale bars are 50 nm in (A, B). (C) Plots showing the number of prefusion (black) and postfusion (red) S trimers per virion against the virion surface area for each of 5 variants. Each data point represents a single virion from 3D tomograms ($n = 15$). (D) Summary of the total S per virion versus S per unit surface area (per 1000 nm$^2$). The Greek letters represent the variant names as in (A). B.1′ indicates the results from an independent preparation described in our previous study (Ke et al, 2020). The dot represents the mean and the error bars in both directions represent standard deviation for $n = 15$ virus particles. Source data are available online for this figure.

617–644), and FPPR (residues 823–853) regions have structural differences approaching 10 Å Cα RMSD. In addition, the Gamma variant differs in the NTD next to D138Y, in the receptor binding site in RBD (next to E484K and N501Y), and in the HR2 helix bundle, all of which are known to be heterogeneous in S structure. Lastly, the D950N mutation induces greater than 2 Å Cα RMSD structural differences between Delta/Mu and B.1 (Fig. 3B). These divergent regions are discussed in further detail below.

## Structural analysis reveals similarities among the variants in NTD and RBD

Aligning the NTD and RBD domains individually against the B.1 structure reveals that the overall structures are highly similar, despite the presence of multiple mutations in these domains (Fig. 4A,B). The NTD is a site of immune vulnerability for the virus (Voss et al, 2021). An antigenic supersite on the NTD has been described that constitutes the N terminus of NTD (residues 14–20), a β-hairpin (residues 140–158), and a loop (residues 245–264) (McCallum et al, 2021b). This supersite is frequently mutated in SARS-CoV-2 variants, particularly with deletions (McCarthy et al, 2021). The periphery of the NTD is highly dynamic and structurally heterogeneous in the absence of antibodies. Consequently, these flexible regions of our S-trimer structures from virions are not well-resolved and do not offer a structural explanation for phenotypes associated with variation in these regions (Fig. 4A). The global architecture of the RBD is similar among variants and is also consistent with structures from purified S (Fig. 4B). Our results confirm that the mutations in the RBD do not lead to any substantial changes in the overall structure. This implies that the effect of variation in this region is large via direct modulation of antibody and receptor binding through individual amino acid changes rather than any substantial structural rearrangement (Zhu et al, 2021).

## Conformational flexibility of S

To assess the conformational heterogeneity of S, we first performed a B-factor analysis on the residue level. The lower the B-factor, the more rigid the structure is. Our results demonstrate that S2 has a lower B-factor than S1 for all variants, indicating S2 is less structurally flexible (Fig. EV3A–C). The flexibility of S1 reflects the capacity of S to adopt different conformations, including open, closed, and locked states (Ke et al, 2020; Xiong et al, 2020; Toelzer et al, 2020). Notably, Mu variant has a much lower B-factor in the RBD region, relative to its NTD, indicating that the RBD in Mu variant is structurally more rigid.

To further assess the conformational states of the RBD, we performed focused classification on the S1 region (using a mask including the NTD plus the open/closed RBD) (Appendix Fig. S1). For all variants, we identified S trimers in both open and closed states (Appendix Fig. S1; Appendix Table S1). The majority of S

trimers from all strains have all three RBDs in the closed state (hereafter, closed state), ranging from 73% (Mu) to 88% (Delta); the Mu variant has the most populated 1-RBD open state (hereafter, open state), constituting 25% (Appendix Fig. S1). This contrasts with studies of recombinant, purified S ectodomains where the majority of the S trimers are in the open state (at least 1-open RBD), varying from 55 to 100%, depending on the virus strains and how the samples were prepared (Gobeil et al, 2021; Cai et al, 2021; Zhang et al, 2021a). In our study, S trimers are assembled and anchored on the native viral envelope bilayer with all other surrounding viral structural proteins, including M, E, and N. They are also subjected to fixation using paraformaldehyde. Studies from in vitro purified recombinant proteins are either missing the anchor of the membrane, the context of the viral proteins surrounding S, or both, and are typically unfixed. The conformational shift toward the closed state in S on virions may reflect the native context, or could be a result of chemical fixation of the virus preparation.

Apart from open and closed states, previous studies have identified an additional conformation called the 'locked state' when introducing engineered disulfide bonds. In the locked state the NTD and RBD are in a closed position, but more tightly packed than in a typical closed state (Qu et al, 2022). We performed further focused classification on the S1 region (mask on NTD plus closed RBD) to see whether a subpopulation of S trimers is in the locked state. Using this approach, 30% of the S monomers in the Mu variant are in the locked state, while the locked state was not observed for other variants. In the locked state of Mu variant, a 10 Å displacement of the NTD toward the center of the threefold axis was detected (Fig. 4C). This 10 Å shift allowed a newly established salt bridge between K113 (NTD) and E471 (RBD), keeping the RBD and NTD more tightly packed (Fig. 4C). The Mu variant has been reported to be about 10 times more resistant to antibodies elicited by either natural SARS-CoV-2 infection or mRNA vaccination as its parent B.1 strain (Uriu et al, 2021). The prevalence of the locked state might contribute to this behavior by altering the conformational distribution of the antigenically vulnerable NTD (McCallum et al, 2021b). Our data do not provide an obvious structural explanation for the presence of the locked state in the Mu variant.

It was previously reported that the fatty acid linoleic acid is bound into a pocket in the RBD in the locked state and that this may play a role in stabilizing S (Toelzer et al, 2020; Xiong et al, 2020; Qu et al, 2022; Toelzer et al, 2022). This free fatty acid-binding pocket is a conserved hallmark in pathogenic beta-coronavirus S (Toelzer et al, 2022). Linoleic acid is, however, absent from all our resolved structures, including the Mu variant locked conformation (Fig. 4D). This confirms that linoleic acid binding is not required to adopt the locked state. It has also been reported that SARS-CoV-2 recruits the heme metabolite biliverdin to its NTD epitope for immune evasion (Rosa et al, 2021) and

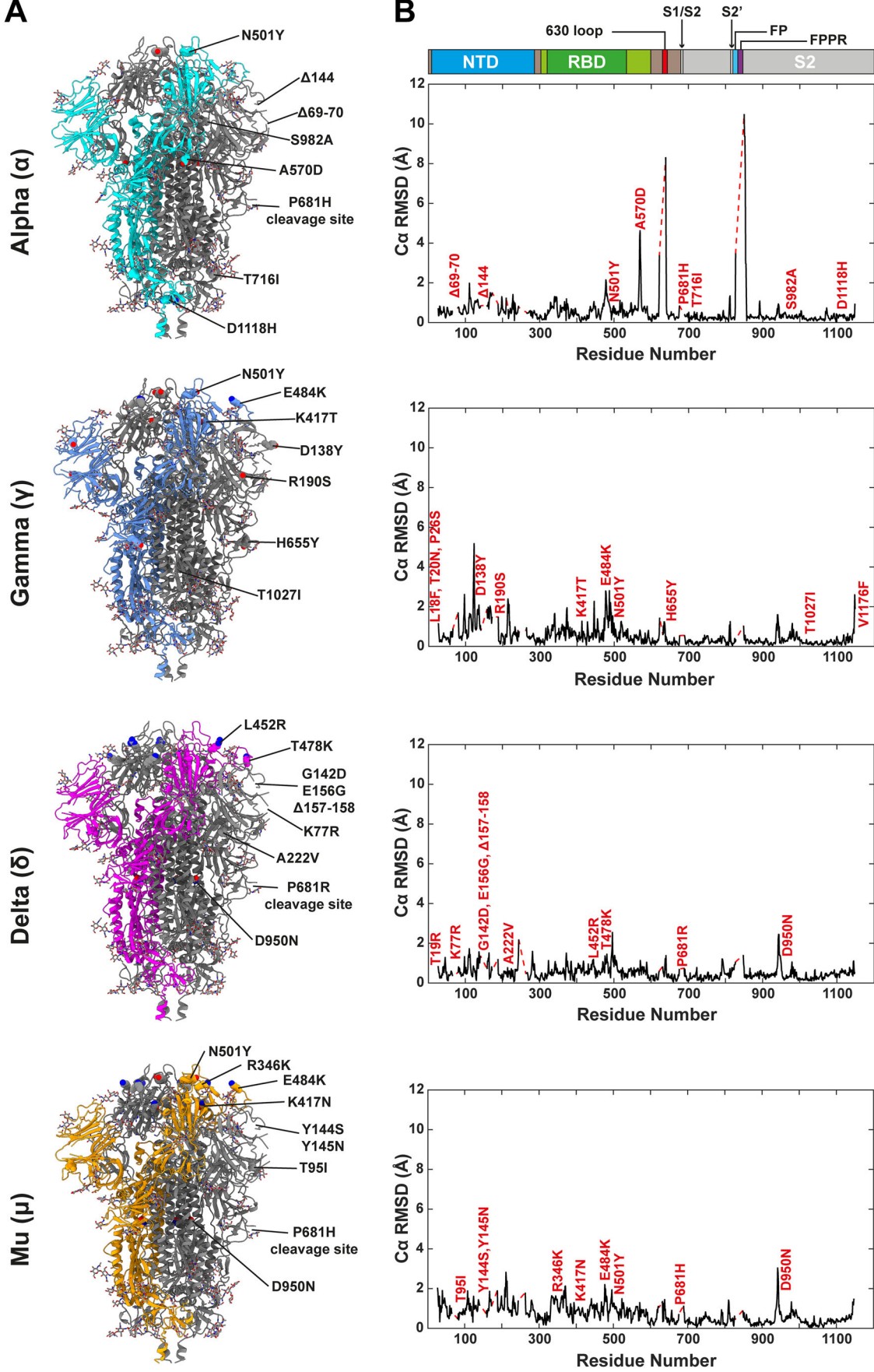

**Figure 3.** Mutation-induced structural variations among variants.

(A) Spike mutations mapped onto the respective variants. A single chain is colored in each model and the mutations are marked with ball representations. (B) The Cα RMSD (Å) between the atomic models of each variant and the B.1 variant (right panels) plotted against residue number. The linear domain architecture of S is included for orientation. See also Fig. EV1. The positions of amino acid variations are marked on the Cα RMSD plots. Unmodelled gaps are indicated by red dashed lines.

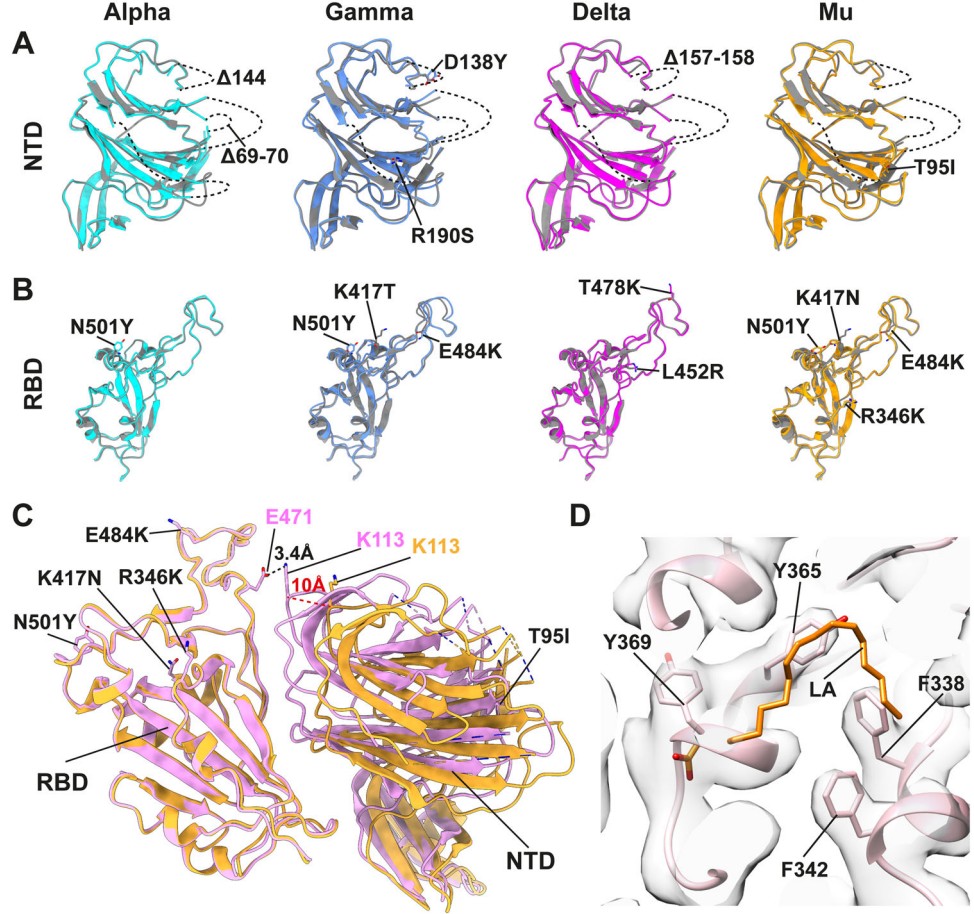

**Figure 4.** Structural comparison of the NTD and RBD between B.1 and variants.

(A, B) NTD and RBD from each variant were locally aligned to B.1 (gray color). Amino acid differences between variants are labeled. Dashed lines represent unresolved regions, indicating structural heterogeneity. Key mutations are labeled and represented as stick model. (C) In the Mu variant, the NTD (right domain), is found in two different positions relative to the RBD (left domain). One position represents the closed state (orange) and another the locked state (pink). In the locked state, the NTD is ~10 Å closer to the RBD than in the closed state, permitting the formation of a salt bridge between K113 and E471. Dashed lines indicate unresolved regions. (D) The locked state of Mu does not contain any density at the previously described binding site of linoleic acid (LA) [PDB 6ZB5].

cryo-EM studies of recombinant, purified S trimers show the presence of biliverdin density in both closed and locked states (Qu et al, 2022). However, biliverdin is absent in all of our on-virus structures, potentially due the serum-free cell culture conditions in which the viruses were propagated. We cannot rule out that the absence of linoleic acid and biliverdin is due to PFA fixation, but it is more likely that they were not incorporated into S on the virions.

## The Alpha variant mutation A570D modulates RBD opening by regulating a secondary structural switch

Our quantitative Cα RMSD analysis identifies particularly large structural differences between Alpha and B.1. The major

differences are around the A570D mutation, in the 630 loop (residues 617–644), and in the FPPR (residues 823–853) (Fig. 3). Multiple studies have suggested that the 630 loop and the FPPR are able to adopt alternate conformations that may modulate RBD opening (Xiong et al, 2020; Cai et al, 2020; Zhang et al, 2021a; Cai et al, 2021; Qu et al, 2022). Studies of purified full-length S proteins in detergent found that 630 loop and FPPR motifs are folded in the closed conformation of B.1, and all unfolded in the closed conformation of Alpha, suggesting that the two motifs could regulate the conformational states of S (Cai et al, 2021).

The on-virus structure of the B.1 closed state is similar to the one we determined previously from BavPat1 B.1 (Ke et al, 2020), (the B.1 strain analyzed here differs by an additional V1068I

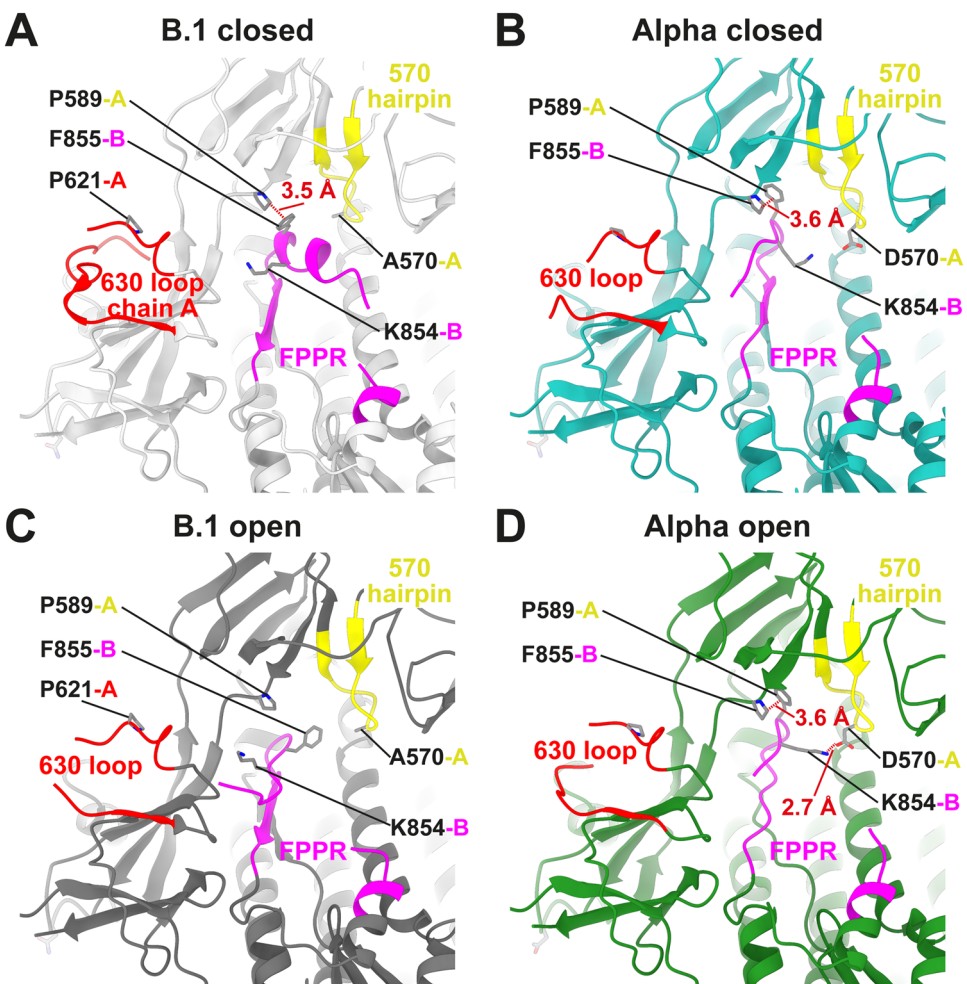

**Figure 5.  Structural changes modulated by A570D.**

Four structures are illustrated: B.1 closed (**A**), Alpha closed (**B**), B.1 open (**C**), chain A is open chain), and Alpha open (**D**), chain A is open chain). Refer also to Figure EV4. The 630 loop (617–644) is in red, FPPR (823–862) is in magenta, 570 hairpin (565–575) is in yellow. Note, the α-helix (848–856) in B.1 closed structure (**A**) is changed to β-strand in Alpha closed structure (**B**). In addition, the A570D mutation leads to a new salt bridge formation between K854 and D570 in the Alpha variant in both closed (**B**) and open (**D**) states.

substitution) (Fig. EV1B). The significantly higher resolution of the structure we present here allows more reliable determination of side-chain orientations and allows more residues in the 630 loop and FPPR to be resolved (Fig. 5A). The structure generally agrees with that determined from recombinantly expressed and purified full-length B.1 S trimers (Zhang et al, 2021a).

The closed form of Alpha has important structural differences to the closed form of B.1. First, residues 852–855 (chain B) which form the end of the final α-helix in the FPPR, instead adopt a β-strand conformation, folding down and packing against residues 858–861 (β-strand) (Fig. 5B). F855, which approaches the outside of P589 (chain A) in the closed B.1 (Fig. 5A), is instead packed against the inside of P589, while K854 (chain B) has rotated from its position facing the 630 loop (chain A) towards the 570 hairpin (chain A) at the base of the RBD (Fig. 5B). The 570 hairpin has moved ~5 Å outwards such that K854 faces D570 at the A570D variant (Alpha) position. The 630 loop is largely disordered (Fig. 5B).

In the B.1 S trimer, upon opening of the RBD in chain A to make the receptor binding site accessible, we observe an outwards movement (~5.7 Å) in the 570 hairpin. The directly neighboring residues 852–855 from chain B are in the β-strand conformation seen in the closed Alpha, but K854 has not rotated and continues to face the 630 loop (Fig. 5C). Instead, residue F855, which packs against the outside of P589 in the closed B.1 and the inside of P589 in the closed Alpha, rotates towards chain A residues 568–572. On the opposite side of the trimer, chain A residues 855–858 have also transitioned to the β-strand conformation, F855 packs against the inner surface of P589, and the adjacent K854 rotates towards the 570 hairpin. This is very similar to the conformation seen in the closed Alpha, but whereas in the closed Alpha K854 appears to interact with the variant A570D, at this position in the open B.1, the 570 hairpin from chain C is still in its closed position and K854 appears to interact with D568 (Fig. EV4G). The end of the FPPR of chain B remains in the helical conformation seen in the closed state, while the 630 loop of chain A, adjacent to 852–855 from chain B, is

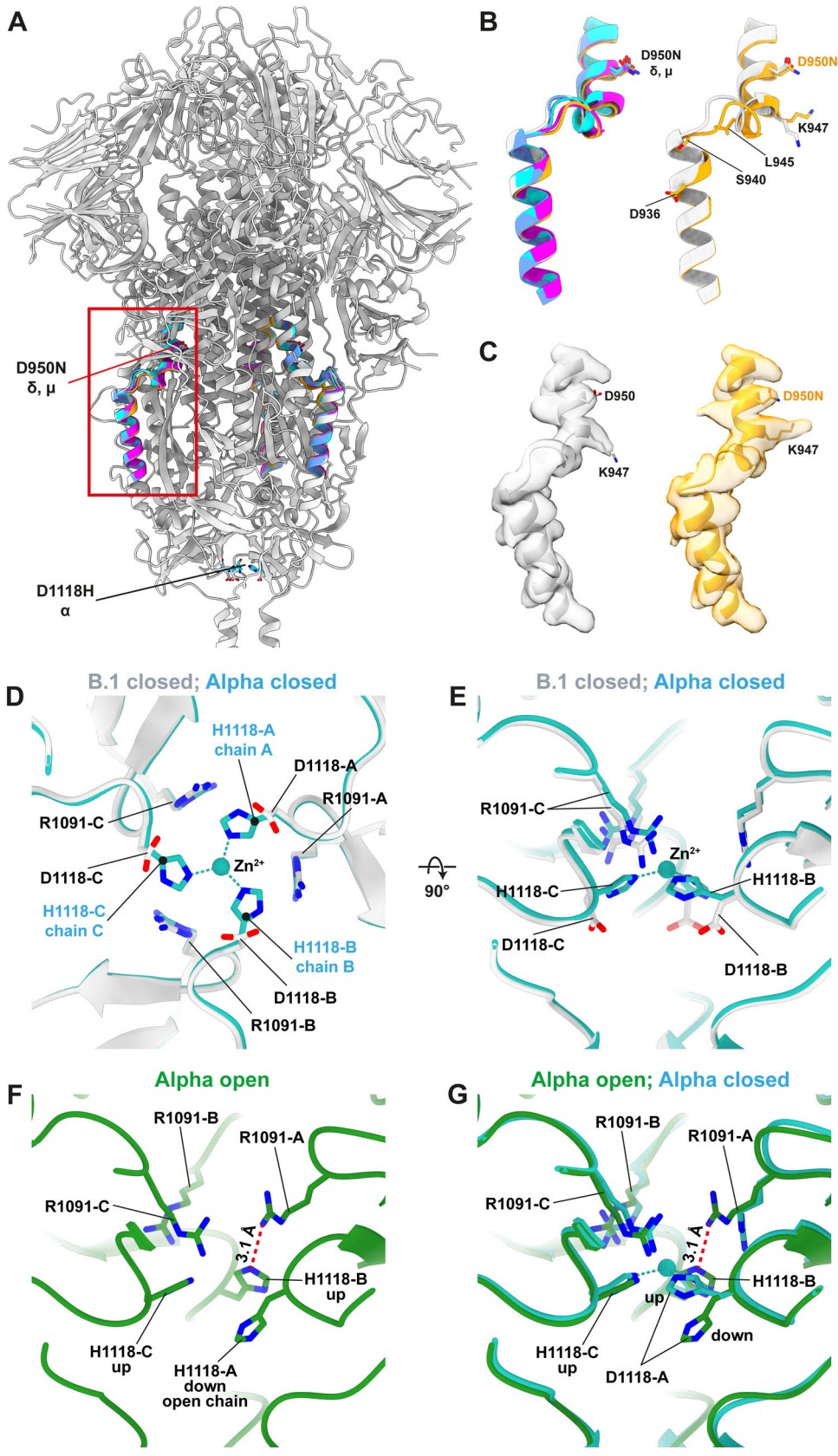

**Figure 6. Structural changes induced by D950N and D1118H.**

(A) Overall structure of B.1 S (light gray) with indicated positions of D950N from Delta (δ) and Mu (μ) and D1118H from Alpha (α). Residues 926–955 are colored for each variant: cyan is Alpha, blue is Gamma, magenta is Delta, and gold is Mu. The red box indicates the location of the region shown in (B, C). (B, left) D950N mutation in Delta (magenta) and Mu (gold) variants induces a structural shift in the 940–945 loop relative to B.1 (light gray), while Alpha (cyan) and Gamma (blue) variants (without D950N mutation) remain structurally similar. The positions of residues 940 and 945 are marked in (B, right). (B, right) D950N mutation and structural comparison between B.1 and Mu. Note 936–940 fusion core helix at HR1 remain unchanged upon D950N mutation while the side-chain rotamer for K947 differs between B.1 and Mu. (C) Model-map density fit for regions shown in (B). Left shows density map for B.1 closed and right is Mu closed. (D–G) Effect of D1118H mutation in Alpha variant. The structures shown are color-coded with the names indicated on the top of each panel. Chain A is the open-RBD chain. (D) Comparison between B.1 and Alpha closed structures. Bottom-up view from the viral membrane. The "upwards" conformation of H1118 in Alpha coordinates an ion, likely to be $Zn^{2+}$, which is absent in B.1. The downwards conformation of H1118 is illustrated in Fig. EV5. (E) 90° tilted view of D. (F) The same view as in (D) but with Alpha open structure illustrated in green. (G) Same view as in (F) showing the predominant conformations of H1118 in the open form of Alpha: down, chain A; up, chains B and C. The closed conformation of Alpha from panel (E) is superimposed in blue for comparison.

largely unfolded. In contrast, the 630 loop of chain C, adjacent to 852–855 from chain A, is folded. The structural transitions involving residues K854 and F855 were not observed in previous studies using recombinant purified full-length B.1 proteins—instead these residues were found in an α-helix in the closed form and in all chains of the open form in recombinant protein (Zhang et al, 2021a).

In the open form of Alpha, residues 852–855 and the 630 loops are generally in the positions observed in the Alpha closed form (Fig. 5D). There is some increased ordering of the 630 loop in chain C, consistent with the relative increase in the order observed for the same residues in the B.1-open form. The 570 hairpin has moved slightly further out in chain A, and less far in chain C, where it is possible that K854 in chain A is interacting with D568 rather than D570, though this cannot be said in confidence (Fig. EV4H).

The above observations suggest that upon opening of the B.1 chain A RBD, the 570 hairpin at the base of the RBD moves slightly outwards (Fig. 5C). This is associated with a helix-to-strand transition of resides 852–855 at the end of the FPPR in the neighboring chain B, where F855 rotates towards the "out-position" 570 hairpin, and in chain A, where K854 rotates towards D568 in the "in-position" 570 hairpin (Fig. 5). We can only speculate as to the functional relevance of these transitions, but suggest that the proximity of K854 and D568 at a position opposite the raised RBD may both stabilize the folding of the 630 loop and modulate the position of the 570 hairpin, either hindering opening of the chain C RBD or favoring it. In the Alpha variant, a plausible model is that the A570D mutation stabilizes an interaction between the rotated K854 and the "intermediate-position" 570 hairpin, thereby allowing these transitions to happen already in the closed conformation of the S prior to RBD opening (Fig. 5). Of note, B.1 which appeared and became dominant early in the pandemic, and from which all variants tested here are descendants of, removed an interaction between K854 and D614, which may also favor these transitions (Zhang et al, 2021a; Ke et al, 2020). In addition, our structures show that residues 852–855 at the end of the FPPR adopt an α-helix conformation in all the other variants (Gamma, Delta, and Mu), indicating that it is the Alpha A570D mutation that favors the helix-to-strand switch (Figs. 5 and EV4; Appendix Fig. S2).

## D950N mutation modulates the structural variation of the fusion core

The D950N mutation induces a structural difference between Delta/Mu (which both contain the D950N mutation) and B.1,

indicated by >2 Å Cα RMSD (Fig. 3). When aligning the HR1 residues, the position of loop 940–945 in Delta and Mu (N950) differs to B.1 and there is a change in the rotamer conformation of K947; these changes are not observed for Alpha and Gamma (D950) (Fig. 6A–C). The fusion core at residues 936–945 in HR1 has been reported to be critical for the conformational change when S transitions from prefusion to postfusion (Korber et al, 2020; Xia et al, 2020) and we speculate that the D950N mutation may modulate this transition (Furusawa et al, 2023) (Fig. 6B,C).

## RBD opening induces long-range allosteric changes at the base of S: D1118H mutation

The Alpha variant has an aspartate to histidine mutation at residue 1118 (D1118H) (Fig. 6A). The histidine is at the base of the trimeric S, facing the 3-fold symmetry axis. In B.1, D1118 is oriented downwards towards the membrane; in Alpha, H1118 has two conformers with one orienting upwards away from the membrane (Fig. 6D,E), and an additional orienting downwards towards the membrane (Fig. EV5). When modeling H1118 upwards away from the membrane, the three histidine residues coordinate an ion, which based on coordination geometry is likely to be $Zn^{2+}$, directly on the threefold symmetry axis (Fig. 6D,E). In previous structures of the Alpha variant, determined from purified recombinant S ectodomain, H1118 has been built in the downward conformation (Gobeil et al, 2021; Cai et al, 2021). Directly above residue 1118 is an arginine at position 1091 oriented towards the membrane. There is a subtle upward movement of R1091 in the Alpha variant (Fig. 6E). In summary, the H1118 mutation (Alpha) permits a stabilizing interaction (i.e., ion coordination) in the closed conformation that the D1118 (B.1) cannot make.

In contrast to the closed conformation, in the open conformation of Alpha, chain A H1118 continues to have two conformers, with a slightly stronger "down" conformation, while H1118 in chain B and chain C are now primarily in the upwards conformation (Fig. 6F,G). The density assigned to $Zn^{2+}$ is present, but weaker, in the open conformation. The downward-facing H1118 (chain A) is accompanied by rotations of R1091 in chains A relative to the closed conformation, while R1091 in chain B has not moved. The correlation between these changes at the base of the S and the opening of RBD at the apex of the S, suggests a long-range allostery within the S trimer. While the complete routes of allostery cannot be derived from our structures, they imply that the raising of the RBD has structural consequences throughout the entire S trimer.

## Concluding remarks: cryo-EM to characterize SARS-CoV-2 evolution

We isolated and cultured a panel of SARS-CoV-2 variants, characterized the S cleavage efficiency (Fig. 1), virion morphology (Fig. 2), and the high-resolution structures of the S trimer directly from authentic virions (Figs. 3–6). Our results indicate that overall virion morphology is largely spherical in all variants—we did not observe invaginated or cylindrical morphologies described by some studies for some strains—suggesting that the evolution of more competitive strains has not led to evolved changes in virion morphology. We observed some variability in the amount of S incorporated into virions in different strains, but the scale of our study cannot rule out that this reflects variation between preparations or within variant lineages.

S trimers undergo a series of structural transitions during assembly, receptor binding, and entry. The range of conformations adopted by S (including cleaved/uncleaved; locked, closed, open; pre- and post-fusion) has evolved to control the transition of S from a locked, to closed, to open state (Qu et al, 2022), and thereby permit transmission under a range of environmental conditions. Structural biologists sample this equilibrium imperfectly: it may be altered by removing the transmembrane domain, by adding an artificial trimerization foldon, by using recombinant expression systems, by subjecting S to biochemical purification protocols, or by using different buffers or detergents. Conformations may also be removed by these processes or by computational classification. Studies of S in situ on virions are also imperfect—the choice of cell lines, as well as methods of inactivation and concentration, might

also alter the conformational equilibrium. Nevertheless, our "on-virus" studies permit structure and conformation to be observed in a close-to-native environment. They have also revealed novel structural details, such as the absence of linoleic acid and biliverdin in the on-virus spike, and the presence of an allosteric route from the RBD to the base of the spike.

As expected, our "on-virus" structures of S confirm that the evolution of strains has not led to any substantial changes in the fold or architecture of S. Instead, they are consistent with three types of functional change in S. First—and not well described by our structures in the absence of antibodies—are local amino acid changes that will modulate the immune response. Second are changes that alter the proteolytic cleavage of S by mutation of the furin site. The third type are changes that alter the range of conformations adopted by S. This third type of change includes the increased structural stability and presence of the locked state in Mu, the subtle change in the fusion core observed as a result of the D950N mutation in Delta and Mu, the substantial and complex changes observed in the RBD-opening structural switch as a result of the A570D mutation in Alpha, and the changes in a putative allosteric pathway from RBD to membrane as a result of D1118H in Alpha. All of these changes can be seen as different evolutionary mechanisms for modulating structural transitions.

We are optimistic that the hybrid pipeline combining cryo-EM and cryo-ET will continue to be a powerful technique to study existing and newly emerged infectious pathogens.

## Methods

**Reagents and Tools Table**

| Reagent/resource | Reference or source | Identifier or catalog number |
| --- | --- | --- |
| **Experimental models** | | |
| Vero cells | Nuvonis Technologies, https://nuvonis.com/vero-cell-banks/ | Nuvonis Technologies serum-free Vero cells, |
| **Antibodies** | | |
| Rabbit anti-SARS spike protein | NOVUS | NB100-56578 |
| Rabbit anti-SARS-CoV-2 nucleocapsid | SinoBiological | 40143-R019 |
| Mouse anti-p24 | Abcam | ab9071 |
| Near infra-red (NIR) secondary antibodies, IRDye® 680RD Goat anti-mouse | Abcam | ab216776 |
| Secondary antibody IRDye® 800CW Goat anti-rabbit | Abcam | ab216773 |
| **Oligonucleotides and other sequence-based reagents** | | |
| QuikChange Lightning Multi Site-Directed Mutagenesis kit | Agilent | 210513 |
| Fluorogenic peptides | Cambridge research biochemicals | Custom made |
| **Chemicals, enzymes, and other reagents** | | |
| Laemmli buffer | Bio-Rad | #1610747 |
| Recombinant furin | New England Biolabs | P8077 |
| 4–20% Mini-PROTEAN TGX protein gel | Bio-Rad | 4561094 |
| 10-nm colloidal gold fiducials | BBI Solutions | #SKU EM.GC10/4 |
| **Software** | | |
| GraphPad Prism software | https://www.graphpad.com | N/A |
| Geneious prime | https://www.geneious.com/ | N/A |

                                      

| Reagent/resource | Reference or source | Identifier or catalog number |
|---|---|---|
| MATLAB | https://www.mathworks.com/products/matlab.html | N/A |
| Chimera | https://www.cgl.ucsf.edu/chimera/ | N/A |
| ChimeraX | https://www.cgl.ucsf.edu/chimerax/ | N/A |
| TEM Tomography 5 software | Thermo Fisher | N/A |
| SerialEM-3.8.0 | Mastronarde, 2005 | N/A |
| IMOD-4.10.30 | Kremer et al, 1996 | N/A |
| RELION-3.1 | Zivanov et al, 2018 | N/A |
| RELION-4 | Zivanov et al, 2022 | N/A |
| EMAN2.2 | Galaz-Montoya et al, 2015 | N/A |
| MotionCor2 | Zheng et al, 2017 | N/A |
| CTFFIND-4.1.13 | Rohou and Grigorieff, 2015 | N/A |
| Topaz | Bepler et al, 2019 | N/A |
| crYOLO | Wagner et al, 2019 | N/A |
| ISOLDE | Croll, 2018 | N/A |
| Phenix | Afonine et al, 2018 | N/A |
| **Other** | | |
| Odyssey Imaging System | LI-COR Biosciences | N/A |
| FLUOstar Omega plate reader | BMG Labtech | N/A |
| Titan Krios Transmission Electron Microscope | Thermo Fisher | N/A |
| K3 direct electron detector with a Gatan BioQuantum | Gatan | N/A |
| Falcon 4 camera with Selectris X | Thermo Fisher | N/A |
| C-Flat 2/2 3C holey carbon grids copper | Protochips | N/A |

## Biosafety

All infectious virus work was approved by the local genetic manipulation safety committee of Imperial College London, St. Mary's Campus (center number GM77), and the Health and Safety Executive of the United Kingdom, under reference CBA1.77.20.1.

## Cells and virus

Vero cells (Nuvonis technologies) were cultured in OptiPRO SFM (Life Technologies) containing 2X GlutaMAX (Gibco). The SARS-CoV-2 isolates are from UKHSA except Delta from Sheffield (see acknowledgement). The variants can be accessed under GISAID ID: B.1 (B.1.238, EPI_ISL_475572), Alpha (B.1.1.7, EPI_ISL_723001) (Brown et al, 2021), Beta (B.1.351, EPI_ISL_770441), Gamma (P.1, EPI_ISL_2080492), Delta (B.1.671.2, EPI_ISL_1731019), Kappa (B.1.671.1, EPI_ISL_2742167), and Mu (B.1.621). To produce SARS-CoV-2 virions, Vero cells seeded on T175 cm² flasks were infected with SARS-CoV-2 (P3) at MOI of 0.5. Culture media from infected cells were harvested at 72 h post infection, clarified by centrifugation at $1000 \times g$ for 10 min, fixed with 4% PFA for 30 min at RT and then cleared through a 0.45-μm nitrocellulose filter. To obtain SARS-CoV-2 virions at high concentration, the concentration of fixation-inactivated virions from media were spun down by ultracentrifugation through a 20% (wt/wt) sucrose cushion (120 min at 27,000 rpm in a Beckman SW32Ti rotor; Beckman Coulter Life Sciences). Pelleted particles were resuspended in PBS and stored in aliquots at −80 °C.

## Western blot

For western blot analysis of virions, clarified virus-containing supernatants were serially concentrated by centrifugation through an Amicon® Ultra-15 Centrifugal Filter Unit followed by an Amicon® Ultra-0.5 Centrifugal Filter Unit with 50 kDa exclusion size. The concentrated virus was then inactivated and proteins denatured by heating with 4× Laemmli sample buffer (Bio-Rad) with 10% β-mercaptoethanol and subsequently run on a 4–20% Mini-PROTEAN TGX protein gel (Bio-Rad). After semi-dry transfer onto nitrocellulose membrane, membranes were probed with rabbit anti-SARS spike protein (diluted 1:2000; NOVUS; NB100-56578) or rabbit anti-SARS-CoV-2 nucleocapsid (diluted 1:4000; SinoBiological; 40143-R019). The near infra-red (NIR) secondary antibody IRDye® 800CW Goat anti-rabbit (diluted 1:10,000; abcam; ab216773) was subsequently used to probe membranes. Western blots were visualized using an Odyssey Imaging System (LI-COR Biosciences).

## Pseudovirus cleavage patterns

Pseudovirus-based spike cleavage assays were performed similarly to those described previously (Peacock et al, 2021a). Variant plasmid was generated from a previously described codon-optimized SARS-CoV-2 spike plasmid (Wuhan-Hu-1), using the QuikChange Lightning Multi Site-Directed Mutagenesis kit (Agilent). Pseudovirus was generated and concentrated as previously

described (Peacock et al, 2021a). All spike expression plasmids used in this study contain D614G and K1255*STOP (that results in deletion of the C-terminal cytoplasmic tail of spike containing the endoplasmic retention signal, aka the Δ19 spike truncation).

Pseudovirus was concentrated by ultracentrifugation as described before (Peacock et al, 2021a). Pseudovirus concentrates were lysed in 4x Laemmli buffer (Bio-rad) with 10% β-mercaptoethanol and run on SDS-PAGE gels. After semi-dry transfer onto nitrocellulose membrane, samples were probed with mouse anti-p24 (abcam; ab9071) and rabbit anti-SARS spike protein (NOVUS; NB100-56578). Near infra-red (NIR) secondary antibodies, IRDye® 680RD Goat anti-mouse (abcam; ab216776) and IRDye® 800CW Goat anti-rabbit (abcam; ab216773) were subsequently used to probe membranes. Western blots were visualized using an Odyssey DlX Imaging System (LI-COR Biosciences).

## Peptide cleavage assays

The peptide cleavage assay was adapted from the protocol by Jaimes et al (Jaimes et al, 2019, 2020). Briefly, fluorogenic peptides were synthesized (Cambridge Research Biochemicals) with the sequences TNSPRRARSVA (B.1), TNSRRRARSVA (P681R), TNSHRRARSVA (P681H) and TNSPSLARSVA (monoCS) and were N-terminally conjugated with the fluorophore 5-Carboxyfluorescein (FAM) and the C-terminal quencher 2,4-Dinitrophenyl. Each peptide was tested for its ability to be cleaved by recombinant furin (10 U/mL; NEB; P8077) in a buffer of 100 mM HEPES, 0.5% Triton X-100, 1 mM $CaCl_2$, 1 mM β-mercaptoethanol, pH 7.5. Assays were performed in triplicate at 30 °C and fluorescence intensity was measured at wavelengths of 485 nm and 540 nm every 1 min for 1 h using a FLUOstar Omega plate reader (BMG Labtech). Maximum enzymatic activity (Vmax) was then calculated (Fig. 1G).

## Phylogenetic tree analysis

Phylogenetic tree analysis was performed on the matched genome sequences of the viruses used in this study. Sequences were aligned using Geneious prime (2019), and then a phylogenetic tree was generated using the neighbor-joining method (Fig. EV1C).

## Cryo-ET sample preparation and data collection

Concentrated virus particles were mixed with 10-nm colloidal gold (in PBS solution) in 10:1 ratio. Then 3 μl of the solution was added to a glow-discharged holey carbon copper grid (C-Flat 2/2, 300 mesh, Protochips). Grids were plunge frozen into liquid ethane by back-side blotting using a Leica GP2 cryo plunger (Leica) and stored in liquid nitrogen until imaging.

Cryo-ET data collection was performed essentially as described previously (Ke et al, 2020). B.1 and Alpha variants were collected at Thermo Fisher Scientific (TFS), Eindhoven, the Netherlands. Cryo-grids were loaded into a Titan Krios G4 transmission electron microscope equipped with a cold field emission gun (CFEG) operating at 300 keV and images were recorded on a Falcon 4 camera with a Selectris X post-column energy filter at a defocus range of −1 μm to −3.5 μm. Gamma, Delta, and Mu variants were collected at Max Planck Institute of Biochemistry, Martinsried,

Germany. Data were collected with Thermo Fisher Scientific Titan Krios G3i transmission electron microscope equipped with a field emission gun (XFEG) operating at 300 keV, and images were recorded on a Falcon 4 camera with a Selectris X post-column energy filter at a defocus range of −4 μm to −6 μm. All data were recorded in electron-event representation (EER) format with a 10 eV energy slit in zero-loss mode. Tomographic tilt series between −60° and +60° were collected using TEM Tomography 5 software (Thermo Fisher Scientific) in a dose-symmetric scheme (Hagen et al, 2017) with a 3-degree angular increment starting at 0 degree tilt angle. A total dose of 123 e⁻/Å² per tilt series was distributed evenly among 41 tilt images. The calibrated pixel sizes on the specimen level are 1.179 Å for B.1 and Alpha variants and 1.183 Å for Gamma, Delta, and Mu variants. The number of tilt series collected for each variant are: 61 (B.1), 39 (Alpha), 15 (Gamma), 15 (Delta), and 10 (Mu).

For image processing of the tilt series, EER raw frames for each tilt image were fractionated to create 10 dose fractions consisting 3 e⁻/Å² dose and motion-corrected in RELION-3.1 (Zivanov et al, 2018) and RELION-4 (Zivanov et al, 2022). The dose-filtered images were saved as output in RELION. Tilt angle information was extracted according to the file name from TEM Tomography software, tilt images were sorted using IMOD *newstack* function version 4.10.30 (Kremer et al, 1996). The tilt axis angle was added to the tilt series using IMOD *alterheader* function. For B.1 and Alpha variant, the 10-nm gold fiducials were used for tilt series alignment in IMOD *etomo* (Kremer et al, 1996); for Gamma, Delta, and Mu variants, AreTomo software (Zheng et al, 2022) was used for tilt series alignment without refining tilt axis, but with local alignment (-Patch 5 5) feature such that higher quality of tomograms can be generated. Tomograms were low-pass filtered to 50 Å for better visualization with EMAN2.2 (Galaz-Montoya et al, 2015) and tomographic slices were visualized with IMOD. See Fig. 2 for representative cryo-ET images.

## Cryo-EM sample preparation and single-particle data collection

For single-particle cryo-EM, the same batches of virus solution were frozen on C-Flat 2/2 3C holey carbon grids copper (Protochips) following the same procedure as for cryo-ET, but without adding 10-nm gold fiducials. Datasets were collected at different locations and are summarized in Appendix Table S1.

For B.1 and Alpha variants, data were collected on Krios3 at the Medical Research Council Laboratory of Molecular Biology (MRC-LMB), Cambridge, UK. Cryo-grids were loaded onto a Thermo Fisher Scientific Titan Krios G3 transmission electron microscope with a field emission gun (XFEG) operating at 300 keV. Images were recorded using a Gatan K3 direct electron detector with a Gatan BioQuantum post-column energy filter and a 20 eV energy slit. Movies were saved as TIFF format and dose-fractionated to 48 frames with an accumulated dose of 50 e⁻/Å² in counting mode with SerialEM-3.8.0 (Mastronarde, 2005) at a nominal magnification of ×81,000, corresponding to a calibrated pixel size of 1.061 Å/pixel at the specimen level. Detailed data acquisition parameters are summarized in Appendix Table S1.

A dataset for Gamma as well as a repeated dataset for B.1 were collected at the Thermo Fisher Scientific (TFS), Eindhoven, the Netherlands. Titan Krios G4 transmission electron microscope

equipped with CFEG was operated at 300 kV, images were recorded on a Falcon 4 camera with Selectris X post-column energy filter and a 10 eV energy slit. Movies were saved as EER format and an accumulated dose of 40 e$^-$/Å$^2$ were acquired using EPU software at a nominal magnification of 165,000 X, corresponding to a calibrated pixel size of 0.727 Å at the specimen level. Detailed data acquisition parameters are summarized in Appendix Table S1.

Delta and Mu datasets were collected at the Max Planck Institute of Biochemistry (MPIB), Martinsried, Germany. Titan Krios G3i transmission electron microscope equipped with XFEG was operated at 300 kV, images were recorded on a Falcon 4 camera with Selectris X post-column energy filter and a 10 eV energy slit. Movies were saved as EER format and an accumulated dose of 40 e$^-$/Å$^2$ were acquired using EPU software at a nominal magnification of ×130,000, corresponding to a calibrated pixel size of 0.93 Å at the specimen level. Detailed data acquisition parameters are summarized in Appendix Table S1.

## Quantification of virion diameter and S trimers per virion

For 2D measurements, virion diameters were quantified from cryo-EM micrographs of each variant. First, 100 micrographs were randomly selected from micrographs with defocus between -1.5 and -3 μm. The micrographs were binned down by a factor of 4 (IMOD *binvol*) (Kremer et al, 1996) and lowpass-filtered to 40 Å (EMAN2 *e2proc2d.py*) (Tang et al, 2007) for better visualization. Quantification of virion diameter were done using IMOD by adding open contour points on the long and short axis of each complete virion (*n* = 100) and the average was used as the virion diameter. Note, incomplete virions were not considered for quantification. IMOD *imodinfo* was used to extract the virion diameter information from the saved model files.

For 3D measurements, the virus diameter was measured by placing two sets of markers on the long and short axis of each virion from the central slice of the virus using Chimera software (Volume Tracer) (Pettersen et al, 2004). The first 15 virions were selected based on the completeness of the virion in the tomograms. In order to count the S trimers on the viral surface, the prefusion and postfusion states of S trimers from 15 complete virions were visually inspected and manually identified using Chimera software (Volume Tracer) (Pettersen et al, 2004). The virion diameter (d$_{virion}$) and number of S trimers were then derived from Chimera output files (cmm format). The virion surface area was calculated using Area$_{virion}$ = πd$_{virion}^2$, while S trimers per surface area is calculated by diving the virion surface area (Area$_{virion}$) by the total number of S trimers (prefusion + postfusion S). The summary of the quantification is presented in Fig. EV2.

## Single-particle cryo-EM image processing and 3D reconstruction

RELION-3.1 and RELION-4.0 were used for image processing (Zivanov et al, 2018, 2022) essentially as in our previous study (Ke et al, 2020). In order to speed up data processing for motion correction and contrast transfer function (CTF) estimation, the Scheduler functionality was used for fully automated real-time processing during data collection (Xiong et al, 2020; Zivanov et al, 2018). Both Gatan K3 and Falcon 4 movies were motion-corrected and dose-weighted using RELION's implementation of the

MotionCor2 algorithm (Zheng et al, 2017). For Falcon 4 data, raw movies are fractionated to a dose between 0.8 and 0.9 e$^-$/Å$^2$ per fractionation. After motion correction, non-dose-weighted sums were used to estimate the CTF with CTFFIND-4.1.13 (Rohou and Grigorieff, 2015). S trimers protruding from the virions were automatically picked using Topaz (Bepler et al, 2019) (B.1, Alpha, and Gamma datasets) or crYOLO (Delta and Mu datasets) (Wagner et al, 2019). First, S trimers were manually picked in a random subset of 100 micrographs from each dataset and used for neural network training for Topaz and crYOLO, respectively. Then, we performed Topaz/crYOLO training separately for each dataset due to differences in virus concentration, background noise, image contrast, and types of cameras used. Using the trained and optimized neural networks, S-trimer particles from all micrographs are then automatically picked. To remove bad particles, particles were extracted at pixel size of ~4 Å (bin4 or bin6) and then subjected to 3D classification with alignment (C1 symmetry) using a previously published 30 Å low-pass-filtered S trimer as the initial reference (Ke et al, 2020). Note, 2D classification was not performed and skipped as the viral membrane dominates the 2D alignment. Good classes were selected from 3D classification, re-extracted, and subjected to 3D auto-refinement (C3 symmetry) in RELION. Then, lower-binning particles were extracted and subjected to 3 more rounds of 3D classification with alignment to further remove bad particles. The refined S-trimer positions were subjected to Bayesian polishing to correct for per-particle beam-induced motions with optimal parameters obtained from training on 10k particles. The polished particles were then submitted for another round of 3D auto-refinement, followed by CTF refinement to estimate (anisotropic) magnification, beam tilt (both trefoil and tetrafoil), and per-particle defocus and per-micrograph astigmatism. After each round of Class3D, Bayesian polishing, or CTF refinement, 3D autorefinements (C3 symmetry) were performed, followed by post-processing. The consensus maps for 3-closed RBDs S trimer for each variant were reconstructed and locally filtered. For more details, refer to Appendix Fig. S1 and Appendix Table S1.

To assess the RBD conformational states on S trimers, we performed focused 3D classification on all the variants. Previously, we determined on-virus S-trimer structures using cryo-EM single-particle analysis (Ke et al, 2020). Using a similar pipeline, a focused 3D classification without alignment using symmetry-expanded (C3) (Scheres, 2016) and signal-subtracted particles (Bai et al, 2015) was performed. The mask used for the focused 3D classification was created with Chimera *molmap* command on a single RBD (in both open and closed conformation) plus its neighboring NTD. In each dataset, focused classification on the C3 symmetry-expanded and signal-subtracted monomer particles resulted in an open-RBD class (Ke et al, 2020). These class assignments were then used to recover 3-closed RBDs and 1-open-RBD S trimers. For all strains, a small proportion <2% of 2-open S trimers were assigned in this way. Then, 3D autorefinements were performed for each S-trimer conformation, and standard RELION processing jobs were used for resolution estimation (Post-processing job) and local map sharpening (Local resolution job). We note that the choices of masks and references can influence classification, and also that the initial selection of particles via 3D classification may introduce a bias against flexible open or disordered forms. The absolute numbers of open classes may therefore be underestimated. The pipeline is,

however, the same for all strains so comparative interpretation is valid. For more details, refer to Appendix Fig. S1 and Appendix Table S1.

## Model building and refinement

All model building was performed in ISOLDE (Croll, 2018). The B.1 3-closed RBDs model was rebuilt starting from 7KRQ (Zhang et al, 2021a). One chain was inspected and, where necessary, rebuilt from end-to-end. The geometry of the poorly resolved NTD and RBD was restrained using 6ZGE (Wrobel et al, 2020) and a re-refinement of 6M0J (Lan et al, 2020; Croll et al, 2021), respectively, as reference models. The resulting chain (along with its attached glycans) was used to regenerate the remaining two chains with strict C3 symmetry. After careful checking of the inter-chain interfaces, the result was refined in phenix.real_space_refine (Afonine et al, 2018) with non-crystallographic symmetry restraints, and restraining torsions and atomic positions to those of the starting model. The rebuilt D614G-3-closed model was used as the starting point for the other variants (Alpha, Gamma, Delta, and Mu). Again working first on a single chain, the mutations were applied and the entire chain was checked and rebuilt before imposing C3 symmetry and refining in phenix.real_space_refine.

Each rebuilt 3-closed RBDs model (B.1 and Alpha) was used as the starting point for its corresponding 1-open-RBD model (B.1 and Alpha). First, the internal geometry of the receptor binding domain in the open chain was restrained using a web of adaptive distance restraints (Croll and Read, 2021) before gently tugging it into the open conformation and settling into the local density. Every chain was then systematically checked and rebuilt end-to-end. Final refinement in phenix.real_space_refine used identical settings as for the 3-closed RBDs models, except that symmetry constraints were not used. Finally, all four models were pruned to remove unresolved stretches of protein, and subjected to a final round of refinement in phenix.real_space_refine. Model-map FSCs were generated by phenix.model_vs_map {model file} {map file} {resolution at half-map FSC = 0.143}. More details on model refinement statistics are in Appendix Table S1 and Appendix Figs. S3–5.

## Structural comparison and Cα RMSD calculation

To assess the structural differences at the amino acid level, the alpha carbon (Cα) RMSD was calculated by aligning the structures between each variant and B.1 using Chimera Match Maker and the resulted RMSD Ca was saved as a header file (hdr format). Cα RMSD was then plotted using MATLAB (version 2019b). Where there is a missing amino acid for comparison (either in the reference or the target structure), the Cα RMSD is not assigned and plotted as dashed red lines. The Cα RMSD plot is shown in Fig. 3B.

## Statistical analysis

For comparative statistical analysis, a two-tailed $t$ test was performed in Prism software. As in Fig. 1, the statistical test results ns indicates not statistically significant ($P > 0.05$), asterisk (*) indicates statistical significance at $P < 0.05$ level, ** indicates statistical significance at $P < 0.01$ level, *** indicates statistical significance at $P < 0.001$ level.

## Data availability

The structures are deposited at Electron Microscopy Data Bank (EMDB) under accession codes EMD-45863 (B.1 closed), EMD-45864 (B.1 open), EMD-45865 (Alpha closed), EMD-45866 (Alpha open), EMD-45867 (Gamma closed), EMD-45868 (Delta closed), EMD-45869 (Mu closed). The associated atomic models in the Protein Data Bank (PDB) under accession codes PDB-9CRC (B.1 closed), PDB-9CRD (B.1 open), PDB-9CRE (Alpha closed), PDB-9CRF (Alpha open), PDB-9CRG (Gamma closed), PDB-9CRH (Delta closed), PDB-9CRI (Mu closed).

The source data of this paper are collected in the following database record: biostudies:S-SCDT-10_1038-S44318-024-00303-1.

## Peer review information

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

## Acknowledgements

The authors would like to thank M Whiteley and T de Silva of Sheffield University for the Delta isolate and Angie Lackenby, Shahjahan Miah, Steve Platt, Joanna Ellis, Maria Zambon, Christina Atchison, and Bassam Hallis of UKHSA as well as field staff for the Alpha, Beta, Gamma, Kappa and Mu isolates. The authors thank the staff of the MRC-LMB EM Facility, in particular A Yeates, G Sharov, G Cannone, and S. Chen for supporting the cryo-EM experiments that were performed at MRC-LMB; D Morado for supporting cryo-EM experiments performed at MPI Biochemistry; J Grimmett and T Darling from MRC-LMB and F Beck from Max Planck Institute of Biochemistry for supporting scientific computing; K Qu, Y Shi, F Abid Ali, J Stacey, D Hrebik, H Guo, G Tagiltsev for helpful discussions. Work in the Briggs laboratory was supported by funding from the European Research Council (ERC) under the European Union's Horizon 2020 research and innovation program (ERC-CoG-648432 MEMBRANEFUSION), the Medical Research Council as part of UK Research and Innovation (MC_UP_1201/16) and the Max Planck Society. TPP, JCB, and WSB are funded by UK Research and Innovation to the G2P-UK National Virology Consortium (MR/W005611/1). DHG, CMS, and WSB are supported by the Wellcome Trust grant no. 205100.

## Author contributions

**Zunlong Ke**: Conceptualization; Formal analysis; Validation; Investigation; Visualization; Writing—original draft; Writing—review and editing. **Thomas P Peacock**: Formal analysis; Supervision; Investigation; Visualization; Writing—review and editing. **Jonathan C Brown**: Investigation; Writing—review and editing. **Carol M Sheppard**: Investigation. **Tristan I Croll**: Formal analysis; Writing—review and editing. **Abhay Kotecha**: Investigation; Writing—review and editing. **Daniel H Goldhill**: Investigation; Writing—review and editing. **Wendy S Barclay**: Supervision; Funding acquisition; Project administration. **John A G Briggs**: Conceptualization; Supervision; Funding acquisition; Writing—original draft; Project administration; Writing—review and editing.

Source data underlying figure panels in this paper may have individual authorship assigned. Where available, figure panel/source data authorship is listed in the following database record: biostudies:S-SCDT-10_1038-S44318-024-00303-1.

## Funding

## Disclosure and competing interests statement

The authors declare no competing interests.

# Expanded View Figures

**Figure EV1.   SARS-CoV-2 S mutations and the phylogenetic tree.**

(**A**) S structure side and top views. One of the three chains is color-coded according to the color scheme in (**B**) to illustrate the positions of the structural features that are discussed in the manuscript. (**B**) Mutations in S for the variants investigated in this study. Variants are indicated using both WHO labels (Greek letters) and PANGO lineage nomenclatures. Δ indicates amino acid deletions. Secondary structures are color-coded in the index strain (Wuhan-Hu-1) and the residue numbers are marked in the panel. NTD, N-terminal domain; RBD, receptor binding domain; CTD, C-terminal domain; 630 loop, a loop which contains residues 617–644; S1/S2, furin cleavage site; S2′, S2′ cleavage site; FP, fusion peptide; FPPR, furin peptide proximal region; HR1, heptad repeat 1; CH, central helix; CD, connector domain; HR2, heptad repeat 2. (**C**) Neighbor-joining phylogenetic tree of SARS-CoV-2 variants investigated in this study, together with the Omicron BA.1 variant. Scale bar refers to a phylogenetic distance of nucleotide substitutions per site.

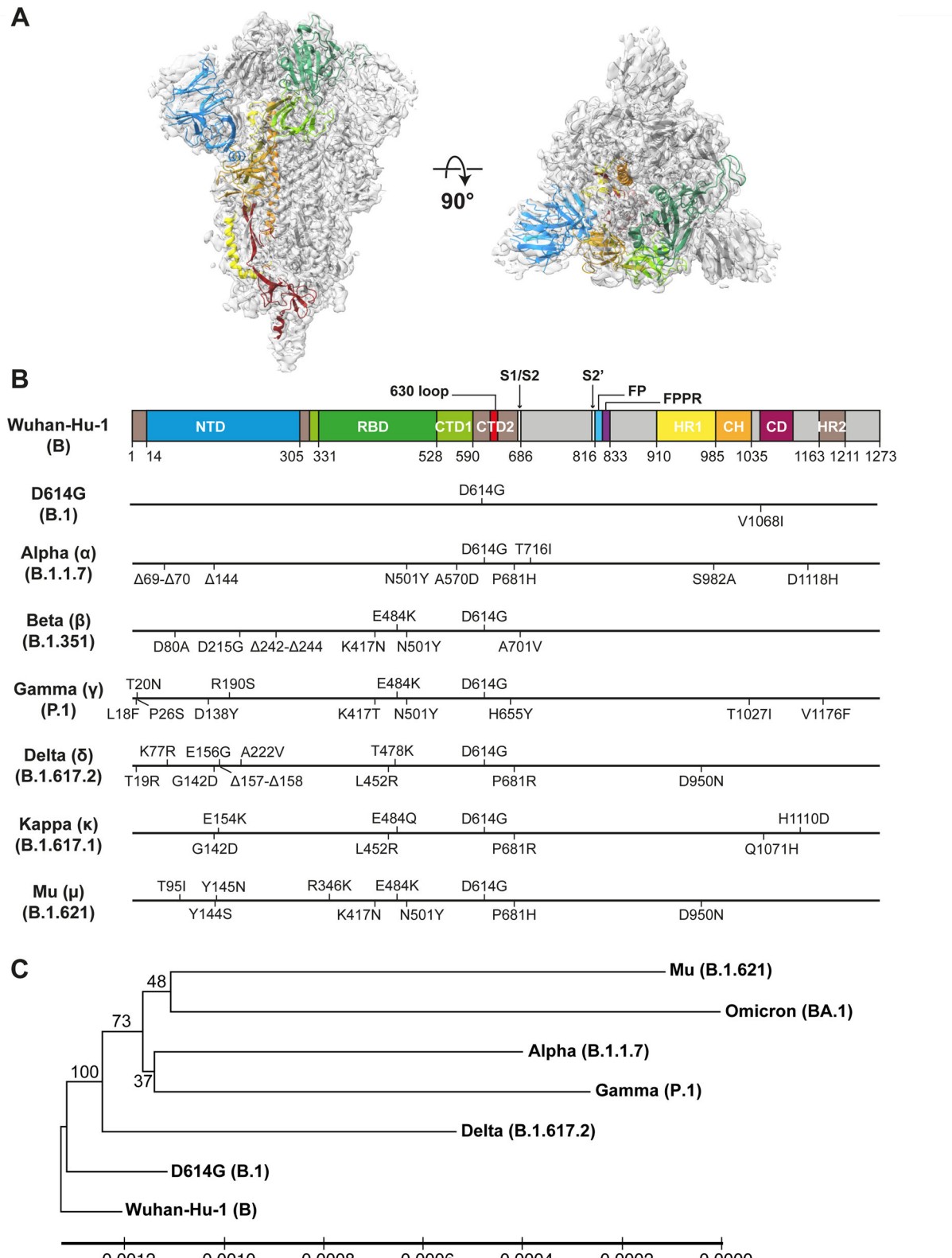

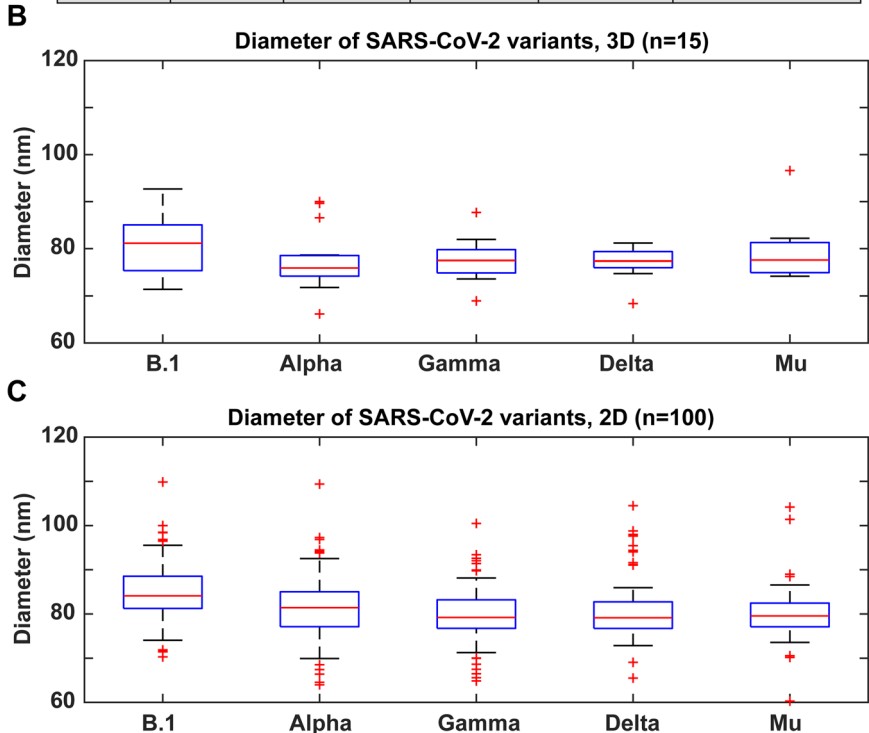

| | prefusion (n = 15) | postfusion (n = 15) | diameter (nm) (3D, n = 15) | diameter (nm) (2D, n = 100) | total S / surface area (per 1000 nm², n = 15) |
|---|---|---|---|---|---|
| B.1 | 20 ± 10 | 1.3 ± 1.5 | 81 ± 7 | 84 ± 7 | 1.0 ± 0.5 |
| Alpha (α) | 23 ± 12 | 1.4 ± 1.6 | 78 ± 7 | 81 ± 9 | 1.3 ± 0.6 |
| Gamma (γ) | 27 ± 13 | 1.5 ± 1.8 | 78 ± 4 | 81 ± 6 | 1.5 ± 0.6 |
| Delta (δ) | 34 ± 14 | 2.4 ± 1.9 | 77 ± 3 | 81 ± 7 | 1.9 ± 0.9 |
| Mu (μ) | 35 ± 16 | 2.8 ± 2.2 | 79 ± 6 | 80 ± 5 | 1.9 ± 0.7 |

**Figure EV2.    Virion diameter quantification.**

(A) The table summarizes the mean numbers of prefusion S, postfusion S, virion diameter (in 3D and 2D measurements), and number of S trimers per unit surface area (per 1000 nm²). Values are presented as mean ± SD; The number of virions used for quantification is in the column header. (B, C) Virion diameter measurements from 3D tomographic reconstructions (B) and 2D projections (C). The n indicates number of virions used for quantification. The box plot represents mean ± SD. A comparative statistical analysis between strains was not performed, because we cannot take possible variation between virus preparations into account. On each box, the central mark (red line) indicates the median, and the bottom and top edges of the box indicate the 25th and 75th percentiles, respectively. The whiskers extend to the most extreme data points not considered outliers. Outliers are plotted individually using the '+' marker symbol.

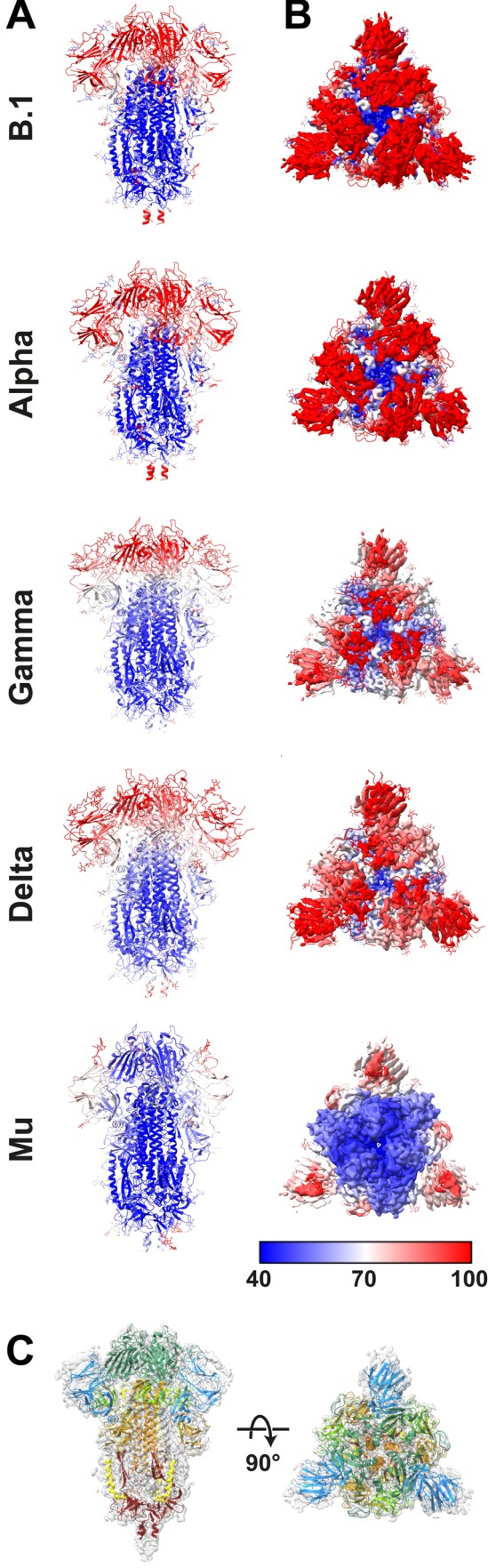

◀

**Figure EV3. B-factor analysis of S structures.**

(A, B) The modeled structures are color-coded according to the residue specific B-factors from 40 Å$^2$ (blue) to 100 Å$^2$ (red). S is shown from the side in cartoon (left column, A) and as a top view with the EM map surface colored by b-factor (right column, B). The lower the B-factor, the more rigid the protein is. In general, the S2 region is mostly rigid across all the variants, indicated by the low B-factor (blue), while the NTD and RBD are more flexible, indicated by the high B-factor (red). Note, that the Mu variant has a relatively rigid RBD while its NTD remains flexible. (C) The secondary structure of S is color-coded according to Fig. EV1.

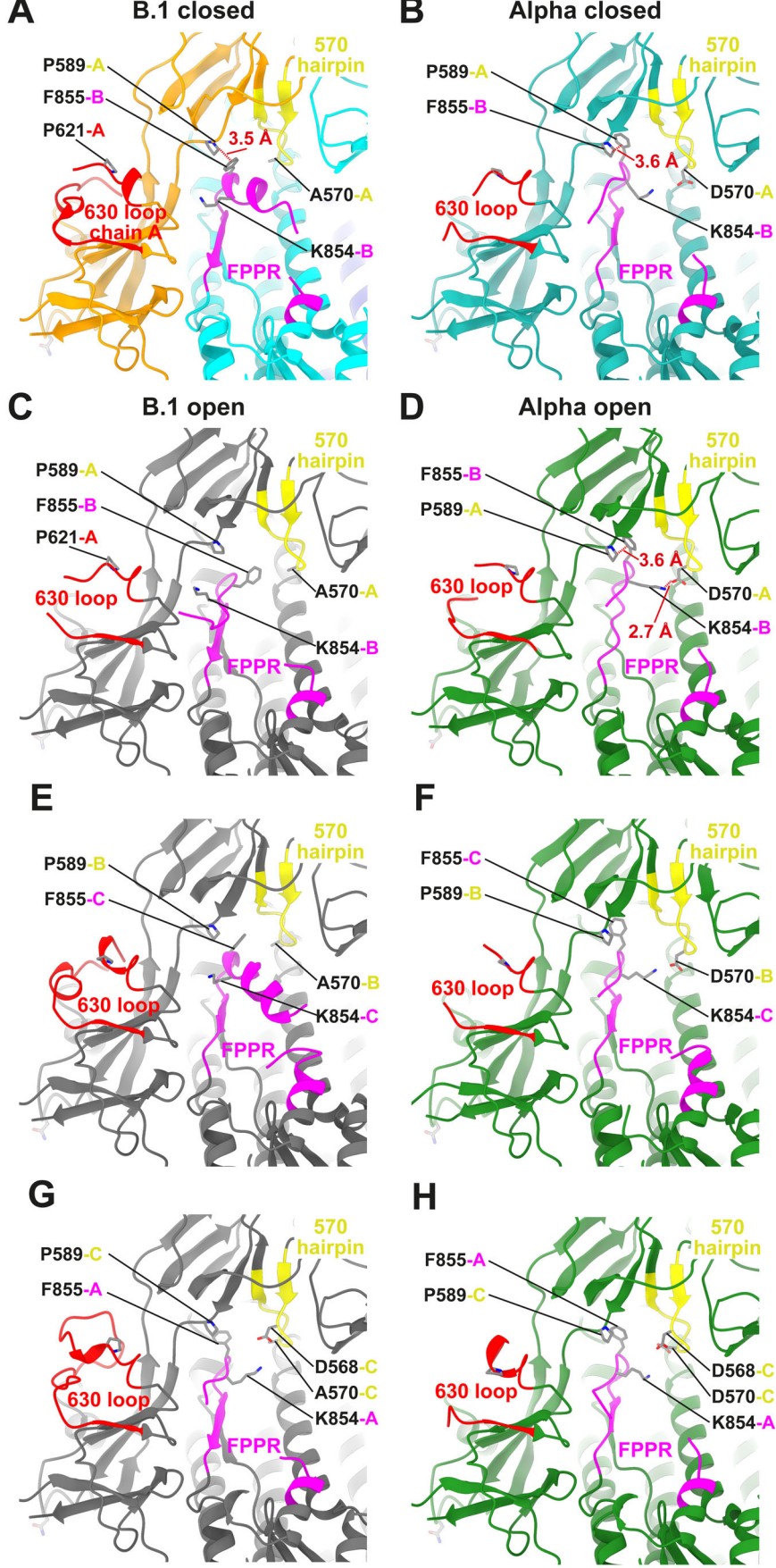

**Figure EV4. Structural changes modulated by A570D illustrated for all chains.**

The three individual chains (ABC) of the 4 structures (B.1 and Alpha, closed and open states) are illustrated here. Color schemes are the same as in Fig. 5. The positions of the three chains are illustrated in (**A**): chain A is in cyan, chain B is in orange, chain C is in blue. (**A–D**) Panels duplicated from Fig. 5 for comparison. (**E–H**) Contacts made by the other chains for B.1 and Alpha open states. The FPPR and interacting residues from each of the three chains from the B.1-open structure are illustrated in (**C, E, G**); the three chains from the Alpha open structure are illustrated in (**D, F, H**).

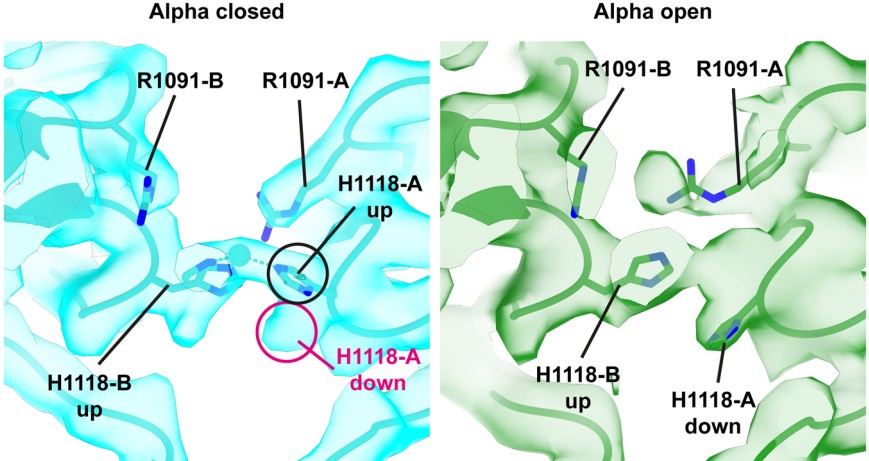

**Figure EV5.  Structural comparison of S structures in the closed and open conformation at mutation D1118H from Alpha variant.**

Left: S structure from Alpha closed state illustrates that H1118 has two conformers, one points upwards (black circle) away from the membrane, and an alternate conformation points downwards (unoccupied magenta circle) towards the membrane. Right: In the open conformation H1118 in chain A (open chain) points primarily downwards and chains B and C points upwards, while R1091 (chain A) has rotated.

