## [Peer Review File · The EMBO Journal]

Virion morphology and on-virus spike protein structures of diverse SARS-CoV-2 variants

Zunlong Ke, Thomas Peacock, Jonathan Brown, Carol Sheppard, Tristan Croll, Abhay Kotecha, Daniel Goldhill, Wendy Barclay, and John Briggs

Corresponding author: John Briggs (briggs@biochem.mpg.de)

Review Timeline:

Transfer Date from Review Commons:	25th Jul 24
Editorial Decision:	16th Sep 24
Revision Received:	7th Oct 24
Editorial Decision:	15th Oct 24
Revision Received:	23rd Oct 24
Accepted:	25th Oct 24

Editor: Ieva Gailite

Transaction Report:

This manuscript was transferred to The EMBO Journal following peer review at Review Commons.

Review #1

1. Evidence, reproducibility and clarity:

Evidence, reproducibility and clarity (Required)

The manuscript by Z. Ke et al. presents the virion morphology and on-virus S trimer structures of various SARS-CoV-2 variants, including B.1, Alpha, Beta, Gamma, Delta, Kappa, and Mu, using a combination of cryo-ET and cryo-EM. The authors report no changes in virion morphology, but they observe that alterations in amino acids at different positions in the spike protein lead to changes in the local structure. Their findings align with previous models, suggesting that these modifications can affect the conformational dynamics of the spike. While the overall research topic is interesting, there are several technique aspects that the authors need to address:

1. It has been previously reported that linoleic acid (LA) binding stabilizes a locked S conformation, resulting in reduced ACE2 interaction in vitro (C. Toelzer et al., *Science* 370, 725-730, 2020). Typically, if there is more locked closed state, the S trimer presents a less open state (C. Xu et al., *Sci. Adv.* 2021; 7: eabe5575). In this study, the authors note that the Mu strain is the only strain that presents the locked state (30% of the S monomers) even though no linoleic acid detected in its structure, however this strain has the least closed state (73%). Is there any structural element that induces this? They should discuss this observation.
2. For the cryo-EM reconstruction of the S trimer structures, the authors report only one type of opening conformation with just one RBD up. It remains unclear whether there exist double or triple RBD-up open conformations in their on-virus S trimer system. This information is important to determine whether the multiple RBD-up open conformation is due to the recombinant condition, or if there is only one RBD up conformation in more endogenous conditions as in the present study. The authors should clarify whether they performed further focused classification on different RBD locations, or they can map back the up RBDs from the symmetry expansion reconstruction to the original S trimer to address this question.
3. It is recommended the authors present the 2D class averages, Euler angle distribution plot, FSC curve, and high-resolution structural features for each of their S trimer maps. Related to this, the workflow display for the S trimer reconstruction in Supplementary Fig. 3 is too compact and small for clear visualization. It is suggested that the authors present these displays in separate Supp figures for improved clarity.
4. In the last paragraph on P. 2, the list of references for available literature for each SARS-CoV-2 variant is rather incomplete. The authors should make the list more complete. For

instance, when discussing the Omicron structure, they should also cite the early two representative works from Eric Xu lab (W. Yin et al., Science 375, 1048-1053, 2022), and Yao Cong lab (Q. Hong et al., Nature 604, 546-552, 2022).

5. The authors state that "As expected, the wild type peptide was efficiently cleaved by recombinant furin while the monobasic mutant was not [Figure 1g]. Consistent with previous publications, P681R showed a significant enhancement of furin cleavage while the P681H mutant did not. This is in contrast to our previous pseudovirus data which suggested P681H alone enhanced cleavage." Could the authors please provide an explanation for this discrepancy?

6. On P. 7, the authors describe that for the on-virus structure of the B.1 closed state spike, "The significantly higher resolution of the structure we present here allows more reliable determination of side-chain orientations and allows more residues in the 630 loop and FPPR to be resolved [Figure 5a]." For the key structural element/side chain origination changes that the authors specifically analyzed, such as those in Fig. 5 and Fig. 6, could they also display the model-map fitting details?

7. For all the detailed interaction interface analysis (such as those displayed in Fig. 5 and Fig. 6), could the authors describe which software or webserver was used to deduce the results? It would also be helpful if they could list the analysis results as supplementary tables.

****Minor points:****

1. In Fig. 5, it is difficult to distinguish the chain A/B/C. Could the authors consider displaying them in different colors or color tones for clarity?
2. In Fig. 6a, it is hard to distinguish the variation portions. Could the authors display the main frames of the S trimer using light grey instead of the currently used dark grey? Also, in this figure, could they indicate the 940-945 loop, and the 936-940 fusion core helix?
3. Fig. 5 and Supplementary Fig. 5 contain redundant content. It is suggested that the authors consider using Supplementary Fig. 5 to replace Fig. 5

2. Significance:

Significance (Required)

The manuscript by Z. Ke et al. presents the virion morphology and on-virus S trimer structures of various SARS-CoV-2 variants, including B.1, Alpha, Beta, Gamma, Delta, Kappa, and Mu, using a combination of cryo-ET and cryo-EM. The authors report no changes in virion morphology, but they observe that alterations in amino acids at different positions in the spike protein lead to changes in the local structure. Their findings are align

with previous models, suggesting that these modifications can affect the conformational dynamics of the spike. While the overall research topic is interesting, there are several technique aspects that the authors need to address

3. How much time do you estimate the authors will need to complete the suggested revisions:

Estimated time to Complete Revisions (Required)

(Decision Recommendation)

Less than 1 month

4. Review Commons values the work of reviewers and encourages them to get credit for their work. Select 'Yes' below to register your reviewing activity at Web of Science Reviewer Recognition Service (formerly Publons); note that the content of your review will not be visible on Web of Science.

No

Review #2

1. Evidence, reproducibility and clarity:

Evidence, reproducibility and clarity (Required)

****Summary:****

Here, the authors utilize cryoET to analyse the structural morphology of SARS-CoV-2 variants. They were able to show that the variants all shared a similar spherical morphology, but differ in in the amount of S incorporated into virions in different strains.

Additionally, the authors utilized single-particle cryoEM on these variants to determine native structures of S on the virus surface in these variants and analyse differences this variation caused in S protein structure. These on-virus structures have revealed evolutionary changes in a near-native environment. This work has revealed a native locked state in the RBD domain previously only seen in engineered S proteins, and that the mutations in the RBD appear to allow evasion of immune response through direct modulation of antibody-S interactions as opposed to structure rearrangement.

****Major comments:****

- Are the claims and the conclusions supported by the data or do they require additional experiments or analyses to support them?

Yes, the data support their claims.

- Please request additional experiments only if they are essential for the conclusions. Alternatively, ask the authors to qualify their claims as preliminary or speculative, or to remove them altogether.

The authors are clear in the shortcomings in their techniques, and do not overclaim their results. They are providing a clear pipeline for their research, and how it can be useful in the field to study viruses. It is intended to demonstrate the power of the strategy and its use in studying viruses.

- If you have constructive further reaching suggestions that could significantly improve the study but would open new lines of investigations, please label them as "OPTIONAL".

- Are the suggested experiments realistic in terms of time and resources? It would help if you could add an estimated time investment for substantial experiments.

- Are the data and the methods presented in such a way that they can be reproduced?

This work seems to be quite reproducible. A reasonable amount of time was spent on data collection and covers a good range of SARS-CoV-2 variants.

- Are the experiments adequately replicated and statistical analysis adequate?

This is addressed by the authors, but they only did a single preparation of each virus for structural analysis. This might be common for structural biology. However, it would be interesting to have tried a few different fixation techniques on one of the samples, to see if this affected the viral morphology/S. While they state that overwhelmingly their samples are spherical, with their samples they do see some virions with dented morphology. It would have been good to test several sample preparation techniques to compare their samples to earlier studies and see whether they can reproduce these. They speculate that overall virions are spherical and variations in morphology is related to sample preparation variation. The statistical analysis seems adequate for the results.

****Minor comments:****

- Specific experimental issues that are easily addressable.

- Are prior studies referenced appropriately?
- Are the text and figures clear and accurate?
- Do you have suggestions that would help the authors improve the presentation of their data and conclusions?

In figure S4, the authors refer to S2 as being blue. In the Mu variant, the whole structure has an overall blue coloration due to the B-factor. While minor for those unfamiliar with the structure it may be useful to put a box over the S2 region.

Fig S5: Might be useful to keep E-H labelled as B.1 open and Alpha open above the figures.

2. Significance:

Significance (Required)

General assessment: provide a summary of the strengths and limitations of the study. What are the strongest and most important aspects? What aspects of the study should be improved or could be developed?

This work reinforces the utility of studying on-virus protein structures to reveal more near-native contexts.

One area that would improve this study is analysis on unfixed viruses in case this fixation causes any changes in protein structure or viral morphology. However, given biosafety concerns and limited access to cryoEM microscopes in a correct biosafety level environment at present it isn't necessarily feasible to do this with regularity.

Advance: compare the study to the closest related results in the literature or highlight results reported for the first time to your knowledge; does the study extend the knowledge in the field and in which way? Describe the nature of the advance and the resulting insights (for example: conceptual, technical, clinical, mechanistic, functional,...).

This work enhances our understanding of how these SARS-CoV-2 viruses evolve and provides us with a near-native picture of how these changes affect its structural morphology. It compares several on-virus S protein structures to identify how these variations affect its structure. Additionally, it directly compares side by side these different viral morphologies and shows that the preparation method of the virus sample can affect the morphology of the virus. In this case, all generally shared a spherical morphology, while in some other cases the viruses have indentations.

Audience: describe the type of audience ("specialized", "broad", "basic research", "translational/clinical", etc...) that will be interested or influenced by this research; how will this research be used by others; will it be of interest beyond the specific field?

This work has a broad interest in our structural understanding of how SARS-CoV-2 adapts to evade immune response in the field of virology. Importantly, a pipeline used to determine these structural results has been established. They are of interest in structural biology, drug design, virology, and may be applicable to other systems as well. Particularly in structural virology this establishes a pipeline for analysis of viral spike structures.

Please define your field of expertise with a few keywords to help the authors contextualize your point of view. Indicate if there are any parts of the paper that you do not have sufficient expertise to evaluate.

Structural biology, cryoEM, virology.

3. How much time do you estimate the authors will need to complete the suggested revisions:

Estimated time to Complete Revisions (Required)

(Decision Recommendation)

Between 1 and 3 months

Yes

Review #3

1. Evidence, reproducibility and clarity:

Evidence, reproducibility and clarity (Required)

The manuscript by Ke et al. reports the high resolution in situ structures of S proteins from different SARS-CoV-2 variants. These are meaningful complements to the structures from recombinant S proteins. Overall, the work is of good quality. The major problems are the missing of strong evidence (such as density) that supports the detailed interactions described in the manuscript. Listed below are some minor issues:

Page3: This analysis suggests large variations in furin cleavage efficiency among the variants. Why the data from pseudo virus and authentic virus are not consistent?

Page5: As have been shown in the data, morphology and number of the S are closely related to the procedure of sample preparation (it is well known that S can fall off from the virion). So is the statistic here meaningful? Does the fixation help to keep the S on the virion?

Page6: Reference models and size of the mask may give quite different results of the classifications. The percentages of closed and open S may vary a lot in different classifications.

Page8: "the A570D mutation stabilizes an interaction between the rotated K854 and the "intermediated-position" 570 hairpin", density around the mentioned residues and region should be shown.

Page9: Figure6-c, densities of the mentioned residues should be shown.

Page9: Observation of a different conformation of H1118 in the RBD "up" conformations is interesting. However, it might be just the multiple conformations of residue H1118. Alternative explanation may be provided.

2. Significance:

Significance (Required)

Most of the studies on the S of coronaviruses are based on the purified recombinant ectodomain of the S protein. The structures in situ reported in this work are important complements to the field, which can confirm some of the in vitro results and also provide new insights on the dynamics of the S protein, especially these caused by the mutations in the variants.

3. How much time do you estimate the authors will need to complete the suggested revisions:

Estimated time to Complete Revisions (Required)

(Decision Recommendation)

Less than 1 month

No

Full Revision

Manuscript number: RC-2023-02325

Corresponding author(s): John, Briggs

[Please use this template only if the submitted manuscript should be considered by the affiliate journal as a full revision in response to the points raised by the reviewers.]

*If you wish to submit a preliminary revision with a revision plan, please use our "Revision Plan" template. **It is important to use the appropriate template to clearly inform the editors of your intentions.***

1. General Statements [optional]

This section is optional. Insert here any general statements you wish to make about the goal of the study or about the reviews.

In this revised manuscript, we have addressed all the points made by the Review Commons reviewers. In addition, we have addressed the points made in a publicly-posted peer review. The changes made are detailed below

Point-by-point description of the revisions

This section is mandatory. Please insert a point-by-point reply describing the revisions that were already carried out and included in the transferred manuscript.

Our responses are in blue. This document includes our responses to all three reviewers from Review Commons as well as our response to one public review posted to the site PRereview.org

We thank the reviewers for their constructive and helpful consideration of the manuscript.

REVIEWER COMMENTS

“Virion morphology and on-virus spike protein structures of diverse SARS-CoV-2 variants”

Reviewer #1 (Evidence, reproducibility and clarity (Required)):

The manuscript by Z. Ke et al. presents the virion morphology and on-virus S trimer structures of various SARS-CoV-2 variants, including B.1, Alpha, Beta, Gamma, Delta, Kappa, and Mu, using a combination of cryo-ET and cryo-EM. The authors report no changes in virion morphology, but they observe that alterations in amino acids at different positions in the spike protein lead to changes in the local structure. Their findings align with previous models, suggesting that these modifications can affect the conformational dynamics of the spike. While the overall research topic is interesting, there are several technique aspects that the authors need to address:

1) It has been previously reported that linoleic acid (LA) binding stabilizes a locked S conformation, resulting in reduced ACE2 interaction in vitro (C. Toelzer et al., Science 370, 725-730, 2020). Typically, if there is more locked closed state, the S trimer presents a less open state (C. Xu et al., Sci. Adv. 2021; 7: eabe5575). In this study, the authors note that the Mu strain is the only strain that presents the locked state (30% of the S monomers) even though no linoleic acid detected in its structure, however this strain has the least closed state (73%). Is there any structural element that induces this? They should discuss this observation.

As we stated in the paper, we were unable to identify a structural explanation for the presence of the locked state in Mu but not in other strains: *“Our data does not provide an obvious structural explanation for the presence of the locked state in the Mu variant.”* Regarding the absence of linoleic acid, we state in the manuscript: *“We cannot rule out that the absence of linoleic acid and biliverdin is due to PFA fixation, but it is more likely that they were not incorporated into S on the virions.”* We have edited our discussion of the observation that linoleic acid binding is not required to adopt a locked state: *“It was previously reported that the fatty acid linoleic acid is bound into a pocket in the RBD in the locked state and that this may play a role in stabilizing S^{19,20,42,63}. This free fatty acid-binding pocket is a conserved hallmark in pathogenic beta-coronavirus S⁶³. Linoleic acid is, however, absent from all our resolved structures, including the Mu variant locked conformation [Figure 4d]. This confirms that linoleic acid binding is not required to adopt the locked state”*. We do not feel able to discuss or speculate further on the reasons for this.

2) For the cryo-EM reconstruction of the S trimer structures, the authors report only one type of opening conformation with just one RBD up. It remains unclear whether there exist double or triple RBD-up open conformations in their on-virus S trimer system. This information is important to determine whether the multiple RBD-up open conformation is due to the recombinant condition, or if there is only one RBD up conformation in more endogenous conditions as in the present study. The authors should clarify whether they performed further focused classification on different RBD locations, or they can map back the up RBDs from the symmetry expansion reconstruction to the original S trimer to address this question.

Indeed, we did map back RBDs from the symmetry expanded reconstruction onto the original trimers, and we do find a small number of particles which appear to have more than one open RBD. We have now added a comment to the methods *“In each dataset, focused classification on the C3 symmetry-expanded and signal-subtracted monomer particles resulted in an open-RBD class⁴⁴. These class assignments were then used to recover 3-closed RBDs and 1-open RBD S trimers. For all strains, a small proportion <2% of 2-open S trimers were assigned in this way.”* As described in our response to reviewer 3, although comparisons between strains are valid, the absolute numbers of open forms may be underestimated. This is also described in the methods: *“initial selection of particles via 3D classification may introduce a bias against flexible open or disordered forms. The absolute numbers of open classes may therefore be underestimated. The pipeline is, however, the same for all strains so comparative interpretation is valid”*. In our previous study of the index strain, tomography methods suggested approximately 14% of particles were in the 2-open state.

3) It is recommended the authors present the 2D class averages, Euler angle distribution plot, FSC curve, and high-resolution structural features for each of their S trimer maps. Related to this, the workflow

Full Revision

display for the S trimer reconstruction in Supplementary Fig. 3 is too compact and small for clear visualization. It is suggested that the authors present these displays in separate Supp figures for improved clarity.

As requested, we have provided FSC curves and map-model FSC plots as Supplementary Figure 9.

We have added a representative Euler angle distribution plot to the supplement as Supplementary Figure 10. We note in the legend that these look different to those from typical single particle reconstructions because the particles are all at the edges of virions, and therefore oriented with the symmetry axis perpendicular to the electron beam. Nevertheless, Fourier space is filled in this scenario.

As requested, we added additional illustrations of high-resolution features for each map, and have also marked key residues as requested by reviewer 3. Please refer to Supplementary Figures 6, 7 and 8.

We have deposited the EM maps and PDB models for readers to download and assess, as requested by reviewer 3.

As requested, we have split Supplementary Figure 3 onto separate pages. Please refer to our new Supplementary Figure 3.

We have not added 2D class averages because as indicated in Supplementary Figure 3, our processing pipeline does not include a 2D classification step.

4) In the last paragraph on P. 2, the list of references for available literature for each SARS-CoV-2 variant is rather incomplete. The authors should make the list more complete. For instance, when discussing the Omicron structure, they should also cite the early two representative works from Eric Xu lab (W. Yin et al., Science 375, 1048-1053, 2022), and Yao Cong lab (Q. Hong et al., Nature 604, 546-552, 2022).

As requested, we have added additional references:

Quote: "*cryo-EM has been used to determine structures of recombinant, purified S trimers from emerging variants including Alpha, Beta, and Gamma* ³⁰⁻³³, *as well as Delta, Kappa, and Epsilon* ^{27,34-36} *and, more recently, Omicron* ³⁷⁻⁴⁰."

5) The authors state that "As expected, the wild type peptide was efficiently cleaved by recombinant furin while the monobasic mutant was not [Figure 1g]. Consistent with previous publications, P681R showed a significant enhancement of furin cleavage while the P681H mutant did not. This is in contrast to our previous pseudovirus data which suggested P681H alone enhanced cleavage." Could the authors please provide an explanation for this discrepancy?

We do not know the reason for the discrepancy, which is also present in the cited, previously published literature. We have added additional lines suggesting possible reasons for this difference "*Consistent with previous publications, P681R showed a significant enhancement of furin cleavage while the P681H mutant did not* ^{54,56}. *This is in contrast to our previous pseudovirus data which suggested P681H alone enhanced cleavage* ⁵⁷. *The reason for this discrepancy between the pseudovirus, live virus and peptide data remains unclear, but could be due to peptides not quite capturing the conformation of the S1/S2*

Full Revision

loop, or a difference in cleavage conditions within the peptide cleavage buffer versus the ER-Golgi intermediate compartment of the cell."

6) On P. 7, the authors describe that for the on-virus structure of the B.1 closed state spike, "The significantly higher resolution of the structure we present here allows more reliable determination of side-chain orientations and allows more residues in the 630 loop and FPPR to be resolved [Figure 5a]." For the key structural element/side chain origination changes that the authors specifically analyzed, such as those in Fig. 5 and Fig. 6, could they also display the model-map fitting details?

We have added additional figure and supplementary figure panels to illustrate the fitting of the model into the density (see Figure 6 and Supplementary Figures 6,7,8).

The resubmission is accompanied by the PDB/EMDB validation reports including per-residue fit data and Q scores.

7) For all the detailed interaction interface analysis (such as those displayed in Fig. 5 and Fig. 6), could the authors describe which software or webserver was used to deduce the results? It would also be helpful if they could list the analysis results as supplementary tables.

We used ChimeraX ISOLDE and Phenix for model building "All model building was performed in ISOLDE¹³. After careful checking of the inter-chain interfaces, the result was refined in phenix.real_space_refine¹⁴ with non-crystallographic symmetry restraints, and restraining torsions and atomic positions to those of the starting model." Differences between structures and the B.1 model were quantified using RMSD Carbon alpha (C α) calculated within Chimera (described in the methods subsection "Structural comparison and C α RMSD calculation" and Figure 3b). Interacting residues were identified by visual inspection of the model and map. While there are some (semi-)automated tools available for interaction interface interpretation we don't think they do a much better job compared to detailed human inspection and they have not been used here.

Minor points:

1) In Fig. 5, it is difficult to distinguish the chain A/B/C. Could the authors consider displaying them in different colors or color tones for clarity?

We updated Figure 5 and added a differently coloured panel to Supplementary Figure 5 to illustrate the different chains, and we hope the new figures will better orient the readers.

2) In Fig. 6a, it is hard to distinguish the variation portions. Could the authors display the main frames of the S trimer using light grey instead of the currently used dark grey? Also, in this figure, could they indicate the 940-945 loop, and the 936-940 fusion core helix?

We updated Figure 6a into light grey as suggested by the reviewer. In addition, we have increased the size of 6a, and have boxed the region highlighted in other panels. Residues 936, 940, and 945 are marked and shown in 6b. Please refer to our new Figure 6.

Full Revision

3) Fig. 5 and Supplementary Fig. 5 contain redundant content. It is suggested that the authors consider using Supplementary Fig. 5 to replace Fig. 5

This partial redundancy was intentional, because we did not want to overwhelm the audience with too much information in the main text figures. We have considered this, and prefer to maintain the current split into main and supplementary figure. We made edits to both figures - supplementary Figure 5 now illustrates the different chains.

Reviewer #2 (Evidence, reproducibility and clarity (Required)):

Summary:

Here, the authors utilize cryoET to analyse the structural morphology of SARS-CoV-2 variants. They were able to show that the variants all shared a similar spherical morphology, but differ in in the amount of S incorporated into virions in different strains.

Additionally, the authors utilized single-particle cryo-EM on these variants to determine native structures of S on the virus surface in these variants and analyse differences this variation caused in S protein structure. These on-virus structures have revealed evolutionary changes in a near-native environment. This work has revealed a native locked state in the RBD domain previously only seen in engineered S proteins, and that the mutations in the RBD appear to allow evasion of immune response through direct modulation of antibody-S interactions as opposed to structure rearrangement.

Major comments:

- Are the claims and the conclusions supported by the data or do they require additional experiments or analyses to support them?

Yes, the data support their claims.

- Please request additional experiments only if they are essential for the conclusions. Alternatively, ask the authors to qualify their claims as preliminary or speculative, or to remove them altogether.

The authors are clear in the shortcomings in their techniques, and do not overclaim their results. They are providing a clear pipeline for their research, and how it can be useful in the field to study viruses. It is intended to demonstrate the power of the strategy and its use in studying viruses.

- If you have constructive further reaching suggestions that could significantly improve the study but would open new lines of investigations, please label them as "OPTIONAL".

- Are the suggested experiments realistic in terms of time and resources? It would help if you could add an estimated time investment for substantial experiments.

- Are the data and the methods presented in such a way that they can be reproduced?

This work seems to be quite reproducible. A reasonable amount of time was spent on data collection and covers a good range of SARS-CoV-2 variants.

Full Revision

- Are the experiments adequately replicated and statistical analysis adequate?

This is addressed by the authors, but they only did a single preparation of each virus for structural analysis. This might be common for structural biology. However, it would be interesting to have tried a few different fixation techniques on one of the samples, to see if this affected the viral morphology/S. While they state that overwhelmingly their samples are spherical, with their samples they do see some virions with dented morphology. It would have been good to test several sample preparation techniques to compare their samples to earlier studies and see whether they can reproduce these. They speculate that overall virions are spherical and variations in morphology is related to sample preparation variation. The statistical analysis seems adequate for the results.

We agree that it would be interesting to compare different fixation techniques, and this is something we have considered. But it would be a major project to validate and certify alternative fixation methods, and to ensure that they also deliver good quality virus at appropriate concentrations. For that reason, a comparison of fixation techniques is beyond the scope of this manuscript. A recent preprint (<https://doi.org/10.1101/2023.10.10.561643>) has imaged unfixed virus, but the challenges of working under containment mean that only small datasets could be studied which could not be analyzed at high resolution,

Minor comments:

- Specific experimental issues that are easily addressable.
- Are prior studies referenced appropriately?
- Are the text and figures clear and accurate?
- Do you have suggestions that would help the authors improve the presentation of their data and conclusions?

In figure S4, the authors refer to S2 as being blue. In the Mu variant, the whole structure has an overall blue coloration due to the B-factor. While minor for those unfamiliar with the structure it may be useful to put a box over the S2 region.

We have added a new panel c in Supplementary Figure 4 to orient the reader, adapted from the new panel in Supplementary Figure 1. We have edited the legend to be clear that color only refers to B-factor. Please refer to Supplementary Figures 1 and 4.

Fig S5: Might be useful to keep E-H labelled as B.1 open and Alpha open above the figures.

We added the labels as requested to Supplementary Figure 5.

Reviewer #2 (Significance (Required)):

- General assessment: provide a summary of the strengths and limitations of the study. What are the strongest and most important aspects? What aspects of the study should be improved or could be developed?

Full Revision

This work reinforces the utility of studying on-virus protein structures to reveal more near-native contexts.

One area that would improve this study is analysis on unfixed viruses in case this fixation causes any changes in protein structure or viral morphology. However, given biosafety concerns and limited access to cryoEM microscopes in a correct biosafety level environment at present it isn't necessarily feasible to do this with regularity.

We thank the reviewer for the positive feedback. Imaging unfixed viruses is indeed a challenge, given the biosafety level needed for SARS-CoV-2. The only study we are aware of is currently available as a preprint (<https://www.biorxiv.org/content/10.1101/2023.10.10.561643>) but was limited to analysis of a small dataset. In the future, it will be beneficial for the community to acquire this information to understand the morphology and spike conformations and structures from unfixed virions under their native conditions.

- Advance: compare the study to the closest related results in the literature or highlight results reported for the first time to your knowledge; does the study extend the knowledge in the field and in which way? Describe the nature of the advance and the resulting insights (for example: conceptual, technical, clinical, mechanistic, functional,...).

This work enhances our understanding of how these SARS-CoV-2 viruses evolve and provides us with a near-native picture of how these changes affect its structural morphology. It compares several on-virus S protein structures to identify how these variations affect its structure. Additionally, it directly compares side by side these different viral morphologies and shows that the preparation method of the virus sample can affect the morphology of the virus. In this case, all generally shared a spherical morphology, while in some other cases the viruses have indentations.

- Audience: describe the type of audience ("specialized", "broad", "basic research", "translational/clinical", etc...) that will be interested or influenced by this research; how will this research be used by others; will it be of interest beyond the specific field?

This work has a broad interest in our structural understanding of how SARS-CoV-2 adapts to evade immune response in the field of virology. Importantly, a pipeline used to determine these structural results has been established. They are of interest in structural biology, drug design, virology, and may be applicable to other systems as well. Particularly in structural virology this establishes a pipeline for analysis of viral spike structures.

- Please define your field of expertise with a few keywords to help the authors contextualize your point of view. Indicate if there are any parts of the paper that you do not have sufficient expertise to evaluate.

Structural biology, cryoEM, virology.

Reviewer #3 (Evidence, reproducibility and clarity (Required)):

The manuscript by Ke et al. reports the high resolution in situ structures of S proteins from different

SARS-CoV-2 variants. These are meaningful complements to the structures from recombinant S proteins. Overall, the work is of good quality. The major problems are the missing of strong evidence (such as density) that supports the detailed interactions described in the manuscript. Listed below are some minor issues:

We thank the reviewer for assessing the manuscript and providing the positive feedback. The major problem raised by the reviewer is the lack of strong evidence for the detailed structural interpretation, because we had not included illustrations of the density for these regions. We have now added the missing illustrations of the density maps for the regions we discuss in the manuscript. See updated Figure 6 and Supplementary Figures 6, 7, 8 and responses to specific reviewer comments. We have also deposited our maps and models to EMDB and PDB databases allowing all readers to interpret them directly.

Page3: This analysis suggests large variations in furin cleavage efficiency among the variants. Why the data from pseudo virus and authentic virus are not consistent?

This difference in total magnitude of spike cleavage efficiency between virus and pseudovirus is consistent with previous reports, including our own (Peacock et al., 2021, Nature Microbiology; <https://pubmed.ncbi.nlm.nih.gov/33907312/>). This is likely influenced by a number of factors including the cell type (Vero for live virus vs HEK293T for pseudovirus production), the stresses and amount of spike and other viral proteins within the cells, and in this specific case the presence of a truncated C-terminal tail in the pseudovirus constructs which alters the trafficking of the spike, and therefore the contact the spike has with endogenous proteases, such as furin. We have included an extra line in the manuscript to reflect this: *“Data using variant-matched pseudoviruses showed a strong correlation with the live virus data with pseudoviruses containing P681R and P681H showing a higher amount of cleavage than those without any changes in the S1/S2 site [Figure 1c-d]. Although total magnitudes of cleavage differed between pseudovirus and virus-expressed S, this is consistent with previous observations⁵², and could be due to differences in how S is expressed and the cell types it was expressed in.”*

Page5: As have been shown in the data, morphology and number of the S are closely related to the procedure of sample preparation (it is well known that S can fall off from the virion). So is the statistic here meaningful? Does the fixation help to keep the S on the virion?

We acknowledge that it is possible that fixation stabilizes S on the virion, but that would represent a stabilization of the native state and would not affect our conclusions. It is also possible that fixation alters the conformational distribution, and in the manuscript we commented on the interpretation of conformational shifts in the presence of fixation: *“In our study, S trimers are assembled and anchored on the native viral envelope bilayer with all other surrounding viral structural proteins, including M, E, and N. They are also subjected to fixation using paraformaldehyde. Studies from in vitro purified recombinant proteins are either missing the anchor of the membrane, the context of the viral proteins surrounding S, or both, and are typically unfixed. The conformational shift towards the closed state in S on virions may reflect the native context, or could be a result of chemical fixation of the virus preparation.”*

Page6: Reference models and size of the mask may give quite different results of the classifications. The percentages of closed and open S may vary a lot in different classifications.

Indeed, references and masks can influence classification. Here, to minimize this, we have performed overall alignment of the particles in the same manner for all strains prior to any classification of conformation. We have then classified the RBD state without alignment using a cautious mask that covers both possible positions of the RBD and NTD - this classification step is therefore not driven by any references. The initial selection of particles for refinement via 3D classification will select against any 3-open forms that have become disordered, or postfusion forms. It may introduce a slight bias against open forms. The subsequent alignment without classification does, however, confirm that open forms are included in the dataset. We have added the following comment in the relevant methods section to make these caveats clear *"We note that the choices of masks and references can influence classification, and also that the initial selection of particles via 3D classification may introduce a bias against flexible open or disordered forms. The absolute numbers of open classes may therefore be underestimated. The pipeline is, however, the same for all strains so comparative interpretation is valid."*

Page8: "the A570D mutation stabilizes an interaction between the rotated K854 and the "intermediated-position" 570 hairpin", density around the mentioned residues and region should be shown.

As requested, and as indicated in response to reviewer 1, we have added panels to show the density around the residues. Please refer to Supplementary Figure 7.

Page9: Figure6-c, densities of the mentioned residues should be shown.

As requested, and as indicated in response to reviewer 1, we have added panel 6c to show the density around the residues. Please refer to our updated Figure 6c.

Page9: Observation of a different conformation of H1118 in the RBD "up" conformations is interesting. However, it might be just the multiple conformations of residue H1118. Alternative explanation may be provided.

H1118 can adopt multiple alternative conformations and we have described this in the manuscript subsection with the title "RBD opening induces long-range allosteric changes at the base of S: D1118H mutation" and in Supplementary Figure 6. There are, however, clear differences in the relative densities for these different conformations which indicate that the conformational dynamics (the relative occupancy of the different conformations) changes in the RBD-up state. We have made some edits to this part of the text to make this point more clearly. We updated Supplementary Figure 6 with rotated views to illustrate the density differences.

Reviewer #3 (Significance (Required)):

Most of the studies on the S of coronaviruses are based on the purified recombinant ectodomain of the S protein. The structures in situ reported in this work are important complements to the field, which can confirm some of the in vitro results and also provide new insights on the dynamics of the S protein, especially these caused by the mutations in the variants.

PREreview of "Virion morphology and on-virus spike protein structures of diverse SARS-CoV-2 variants"

Published March 4, 2024 | Version v1

James Fraser, Luisa Vasconcelos, Liyi Cheng, 9 other authors

Research group: HHMI Transparent and Accountable Peer Review Training Pilot

This Zenodo record is a permanently preserved version of a PREreview. You can view the complete PREreview at <https://prereview.org/reviews/10779310>.

The SARS-CoV-2 virus has experienced tremendous selective pressure over the course of the global pandemic with variants of concern emerging that differ in terms of transmissibility, immunogenicity, and other properties. The major goal of this paper is to determine how sequence changes in the major determinant of these properties, the Spike protein, affect structure, abundance, and distribution of the virion. In contrast to most previous studies, which use purified Spike that isn't anchored to the virus surface (or even a mimic of the virus surface or a biological environment), this study uses electron microscopy/tomography to visualize structures in a more native context. The major potential areas of improvement of the paper are the lack of quantitative comparisons to the "meta ensemble" of structures determined by other methods (single particle EM, X-ray, etc) and the limited discussion of the immune evasion properties of the variant structures with regard to antibody binding footprints. It ends with a potential mechanism for how the conformational equilibrium of certain conformations needed for fusion are favored by specific amino acid changes and argues, quite convincingly, that the insights from the in situ structures determined here are less biased in determining such changes. The manuscript therefore succeeds in its major goals - and points to complex interdependencies of different properties in evolution and not a single conformational coordinate as the result of the selective pressure.

We thank these reviewers for taking the time to study our preprint. We have edited the manuscript in response to their comments.

Major points:

- The paper begins with some variation in Furin cleavage and is mostly concerned with structural analysis, but the link between the biochemical properties and the structural/flexibility parameters uncovered is unclear. Furthermore, the acronym FPPR is only defined in the Supplementary Figure 1 and the potential importance of the proximal peptide needs a bit more introduction to enable the reader to follow the arguments in the paper.

We are aware that the structural analysis does not provide an explanation for the biochemical changes in furin cleavage. Nevertheless, we think that a thorough characterization of the strains requires that both kinds of data are presented.

We have now briefly introduced the FPPR and defined the acronym in the introduction: "*Sequential cleavage of S at the S1/S2 site followed by the S2' site within the S2 domain exposes the fusion peptide (FP), which mediates host-viral membrane fusion, allowing entry of the viral genome into the host cell. It has been reported that a highly flexible fusion-peptide proximal region (FPPR), downstream of the FP, contributes to regulation of the conformational states of S* ¹³."

- Figure 2d is very tantalizing and potentially shows a simple property that is changing through time. Given that the caveats of small n are already noted in the manuscript, more speculation about the figure's interpretation could be interesting. It would be good to elaborate in that discussion beyond: "a comparative statistical analysis between strains was not performed, because we cannot take possible variation between virus preparations into account." For example, were there notable differences in the preparation?

The preparation methods were the same in all cases. We have added an additional datapoint to Figure 2d to illustrate that the results obtained here for B.1 are very similar to those we obtained from the very similar strain (but different source laboratory) used in our previous study. While we are reluctant to risk overinterpreting our data, we have added one sentence of speculation on this result: "*The measured density of trimers in B.1 is very similar to that measured from the almost identical strain used in our previous study*⁴⁴ [Figure 2d]. While it is tempting to speculate that more recent strains have evolved higher densities of S on the viral surface, since each variant was only prepared and imaged once in this study, we cannot rule out that these differences reflect variation in incorporation independent of strain."

- Figure 3 focuses on the trimeric structure and identifying key sites - however, it might be useful for colors to be consistent across figures 3a and 3b. This would make the domains easier to structurally identify, and help the reader to associate domains (cleavage site, NTD, RBD, etc.) with the interpretation of the next couple of figures.

To help the reader to associate domains, we have added a new panel to Supplementary Figure 1 and have referred to it in the legend of Figure 3. In the new panel, one monomer is coloured according to the domain colour scheme,

- Figure 4 details the structural comparison among variants at the NTD and RBD. It appears that the authors used the same global structural alignment as in figure 3, which may not be the optimal choice to examine local structural variations because overall domain shifts may interfere with the comparison. In addition, no reference or PDB code was included for the reference structure with linoleic acid.

In figure 4a and 4b, we did not use the global alignment, but used a local alignment of the NTD or the RBD only. As the reviewers indicate, the local alignment is the better choice here. We have edited the legend to make this clearer. "Figure 4. Structural comparison of the NTD and RBD between B.1 and variants. (a-b) NTD and RBD from each variant were locally aligned to B.1 (grey color)"

We have added the PDB code(6ZB5) for the linoleic acid structure to the figure legend.

Minor point:

- We are confused as to whether they use B.1 as WT interchangeably in the manuscript. More precision in word choice would avoid some confusion with the figure 1 legend as comparisons also mention WT. In the legend for figure 1, the stats reference comparisons to WT, but some only have B.1 and not WT labels.

Full Revision

Indeed, we were inconsistent in this nomenclature and have now used B.1 consistently throughout.

- coulombic potential density (or potential density or density) not electron density

Corrected as requested.

- In figure 2c, consider adding a legend that specifies red is prefusion and black is postfusion.

We edited the figure legend according to this suggestion. The updated Figure 2 also includes a key.

In addition to the changes described above, we have corrected some typographical and style errors in the manuscript.

Dear John,

Thank you for submitting your revised Review Commons manuscript to The EMBO Journal. I apologise for the protracted assessment process due to delays in reviewer report submission and my absence from the office at the beginning of September.

Your manuscript has now been seen by two of the original reviewers, who find that their main concerns have now been addressed and broadly recommend publication of the manuscript. I will therefore be happy to accept the manuscript for publication in The EMBO Journal after textual editing as requested by reviewer #3 and reformatting along the guidelines included below and in the attached document.

Please feel free to contact me if you have any further questions regarding this final revision. Please use the link below to upload the revised files.

Thank you for the opportunity to consider your work for publication, and I look forward to receiving your revised manuscript.

With best wishes,

Ieva

- a point-by-point response to the referees' comments, with a detailed description of the changes made (as a word file).
- a word file of the manuscript text.

- individual production quality figure files (one file per figure)

- a complete author checklist, which you can download from our author guidelines

(<https://www.embopress.org/page/journal/14602075/authorguide>).

- Expanded View files (replacing Supplementary Information)

- a Reagents and Tools Table as part of the Methods section, which can be downloaded from our author guidelines

(<https://www.embopress.org/page/journal/14602075/authorguide#structuredmethods>)

We realize that it is difficult to revise to a specific deadline. In the interest of protecting the conceptual advance provided by the work, we recommend a revision within 3 months (15th Dec 2024). Please discuss the revision progress ahead of this time with the editor if you require more time to complete the revisions.

Referee #1:

The authors have addressed most of my questions, and I would recommend acceptance of the manuscript.

Referee #3:

The revised ms has been significantly improved.

However, for the comment "the A570D mutation stabilizes an interaction between the rotated K854 and the "intermediated-position" 570 hairpin". The author showed the density around the mutation site in Figure S7. The figure is of low quality and indeed shows that the region is sort of disordered. Thus, I would suggest that the authors should tone down their interpretations/conclusion.

REVIEWER COMMENTS

“Virion morphology and on-virus spike protein structures of diverse SARS-CoV-2 variants”

Manuscript number: EMBOJ-2024-118576-T

Corresponding author(s): John, Briggs

Referee #1:

The authors have addressed most of my questions, and I would recommend acceptance of the manuscript.

We thank referee #1 for their feedback.

Referee #3:

The revised ms has been significantly improved.

However, for the comment "the A570D mutation stabilizes an interaction between the rotated K854 and the "intermediated-position" 570 hairpin". The author showed the density around the mutation site in Figure S7. The figure is of low quality and indeed shows that the region is sort of disordered. Thus, I would suggest that the authors should tone down their interpretations/conclusion.

As requested, we have toned down the statement in the updated manuscript. It now reads “*In the Alpha variant, a plausible model is that the A570D mutation stabilizes an interaction between the rotated K854 and the “intermediate-position” 570 hairpin, thereby allowing these transitions to happen already in the closed conformation of the S prior to RBD opening [Fig. 5].*”

We thank all the reviewers for their constructive feedback and help in improving the manuscript.

Dear John,

Thank you for submitting a revised version of your manuscript and for addressing most of the remaining editorial points. I am afraid that there remain a couple of formatting aspects as outlined below that still need to be implemented in the manuscript:

1. Please provide up to five keywords.
2. CRediT has replaced the traditional author contributions section because it offers a systematic, machine-readable author contributions format that allows for more effective research assessment. Please remove the Authors Contributions from the manuscript and use the free text boxes beneath each contributing author's name in our online submission system to add specific details on the author's contribution. More information is available in our guide to authors.
3. Please rename "Competing interests" section into "Disclosure and competing interests statement" (further info: <https://www.embopress.org/page/journal/14602075/authorguide#conflictsofinterest>).
4. Currently, there is a "Results and Discussion" section and a "Conclusions" section. We do allow to combine Results and Discussion sections for shorter manuscripts. Therefore, since separating Results and Discussion sections at this stage would be likely challenging, I would suggest removing the "Conclusions" heading and, instead, start the paragraph with "In conclusion, ..."
5. Individual panels of EV figures are not mentioned in the manuscript text. Please add the corresponding callouts.
6. Please add figure legends for the main and EV figures (but not the figures themselves) to the manuscript text file after references.
7. In the "Data Availability" section, please add resolvable links to the structural datasets. More information about the format of this section can be found here: <https://www.embopress.org/page/journal/14602075/authorguide#dataavailability>
8. In the Appendix file, please add page numbers in the table of contents.
9. Our data editors have flagged the following issues in figure legends that need correcting:
 - Please provide the exact p values in the legends of figures 1d, f-g.
 - Please note that in figures 1d, f-g; there is a mismatch between the annotated p values in the figure legend and the annotated p values in the figure file that should be corrected.
 - Please define the box plots in terms of minima, maxima, centre, bounds of box and whiskers, and percentile in the legends of figures EV 2b-c.
 - Please describe the number and nature of replicates in the legend of figure 2d.
 - Please define the red/black arrowheads/dotted circles in the legend of figure 2b.
10. Papers published in The EMBO Journal are accompanied online by a 'Synopsis' to enhance discoverability of the manuscript. It consists of A) a short (1-2 sentences) summary of the findings and their significance, B) 3-4 bullet points highlighting key results and C) a synopsis image that is 550x300-600 pixels large (width x height, jpeg or png format). You can either show a model or key data in the synopsis image. Please note that the image size is rather small and that text needs to be readable at the final size. Please send us this information together with the revised manuscript.

Please feel free to contact me if have any questions regarding this final revision. Thank you again for giving us the chance to consider your manuscript for The EMBO Journal. I look forward to receiving the revised version.

With best wishes,

leva

leva Gailite, PhD
Senior Scientific Editor
The EMBO Journal
Meyerhofstrasse 1
D-69117 Heidelberg
Tel: +4962218891309
i.gailite@embojournal.org

We realize that it is difficult to revise to a specific deadline. In the interest of protecting the conceptual advance provided by the work, we recommend a revision within 3 months (13th Jan 2025). Please discuss the revision progress ahead of this time with the editor if you require more time to complete the revisions.

The authors addressed the remaining editorial issues.

Dear John,

I am pleased to inform you that your manuscript has been accepted for publication in the EMBO Journal. I am now pleased to inform you that your manuscript has been accepted for publication.

Before we forward your manuscript to the publishers, I would like to suggest minor edits in the manuscript abstract and synopsis. I have also written a short blurb that will accompany the title of your manuscript on our website. Please take a look at the text below and in the attached manuscript text file and let me know if any further edits are needed.

Blurb:

Cryo-electron microscopy and tomography reveal structural variation of Spike protein in intact virions from seven SARS-CoV2 variants.

Synopsis:

The evolution of SARS-CoV-2 variants has influenced viral Spike (S) protein structure and cleavage by furin, and has been proposed to modulate virion morphology. This study presents a characterization of virion morphology, spike density, furin cleavage and "on virus" spike structure from a range of SARS-CoV-2 variants analysed in parallel.

- SARS-CoV-2 variants, including B.1, Alpha, Beta, Gamma, Delta, Kappa, and Mu variants were imaged by cryo-electron microscopy and cryo-electron tomography.
- The evolution of more competitive strains has not led to significant changes in the generally spherical morphology of the virions.
- High-resolution structures of spikes on the virion surface reveal the impact of amino acid variation in the S protein on the protein structure and conformational dynamics throughout the spike.

If you have any questions, please do not hesitate to contact the Editorial Office. Thank you for this contribution to The EMBO Journal and congratulations on a nice study!

Best wishes,

Ieva

Rev_Com_number: RC-2023-02325

New_manu_number: EMBOJ-2024-118576R1

Corr_author: Briggs

Title: Virion morphology and on-virus spike protein structures of diverse SARS-CoV-2 variants